# Assessment of a youth, climate empowerment program: Climate READY

Rachel L. Wellman[1], Anne Henderson[1], Ray Coleman[1], Christopher Hill[1], Bradford T. Davey[2]

[1]FAU Pine Jog Environmental Education Center, Florida Atlantic University, West Palm Beach, Florida 33415, United States
[2]Technology for Learning Consortium, Inc., Stuart, Florida 34996, United States

*Correspondence to*: Rachel L. Wellman (wellmanr@fau.edu)

**Abstract.** This article presents an in-depth assessment of a youth, climate empowerment program, called Climate READY – Climate Resilience Education and Action for Dedicated Youth. It was developed by the Florida Atlantic University Pine Jog Environmental Education Center (FAU Pine Jog) and funded by the National Oceanic and Atmospheric Administration (NOAA)
Environmental Literacy Program The program built climate literacy and community resilience through a three-semester dual enrollment program (NOAA-SEC-OED-2020-2006190). Most student participants (~80%) were from Title-1, high schools in low socio-economic communities vulnerable to extreme weather and environmental hazards in Palm Beach County, Florida (see definition in Appendix 1). The main objectives of this program were to:

1.  increase knowledge of South Florida's changing climate systems,

2.  teach and promote environmentally responsible behavior, which results in the stewardship of healthy ecosystems and a reduction in carbon consumption to mitigate future environmental risks, and

3.  empower students to act as agents of change within the community by teaching community members about local climate impacts and resilience strategies for extreme weather events.

Important characteristics of the program included the following:

a)  Students ages 15 to 17 years old registered for the Climate READY Ambassador Institute (Summer Semester 1) built climate knowledge, explored NOAA Science on a Sphere technology, engaged with scientists and resilience experts, developed communication and advocacy skills, and learned about local resilience solutions. At the end of the course these students were given completion certificates and the title "Climate READY Ambassadors" (CRAs).

b)  An After-school Mentorship (Fall Semester 2) component paired new Climate READY Ambassadors with fourth- and
fifth- grade after-school students ages nine to 11 years old to build community resilience awareness through four structured lessons and the creation of storybooks. Lastly,

c)  Community Outreach (Spring Semester 3) provided ways for Climate READY Ambassadors to share local resilience strategies at public events and promoted civic engagement in climate solutions.

Data were collected from all students in the form of pre- and post-assessment questionnaires during the 2022-2023 academic year.
Summative statistics were analyzed for climate science knowledge, self-identity, self-efficacy, and sense of place. Climate READY Ambassadors felt more prepared, confident, and able to communicate within their communities about climate change and many demonstrated a significantly better understanding of climate science concepts. After-school students showed a better understanding of climate change and were able to identify ways to help reduce the effects of climate change. Both groups of students benefitted from the Climate READY Ambassador mentorship, demonstrating learning by doing and learning by storytelling.

## 1 Introduction

The most recent report from the Intergovernmental Panel on Climate Change (IPCC) provides evidence that climate change is certain to unleash serious impacts globally, including an increase in the exposure of coastal areas to natural disasters (IPCC, 2023).

A community's vulnerability to natural disasters is a combination of the exposure to risk that the community faces coupled with the community's available social, economic, political, and institutional resources that allow them to adapt (IPCC, 2023; Southeast Florida Regional Climate 2023 Compact, 2022). To prepare for a future of increasing hazards, communities will need an informed public that is willing to act on decisions at a personal and civic level. How people prepare for, respond to, and cope with natural disasters is linked to community resilience, -in other words, how well a community can "bounce back" when a natural disaster occurs (Ronan and Johnson, 2005). This kind of preparedness requires a minimal level of environmental literacy, by which we mean, the possession of knowledge and understanding of a wide range of environmental concepts, problems, skills, and abilities. A more environmentally literate public makes more informed decisions and is more involved on a community level, which contributes to community resilience. The complexity of climate change science often makes it difficult for a less climate-literate general public to gain a thorough understanding, severely impeding their ability to make informed decisions for themselves, their families, and their communities.

A solution to this lack of understanding is education, particularly in schools. A survey in 2012 by Florida Atlantic University (FAU) revealed that over 67% of respondents in South Florida felt that the causes, consequences, and solutions to climate change should be taught in K-12 (youth ages 5 to 18 years old) classrooms (Lambert et al., 2012). However, the educational community, particularly in the state of Florida in the United States (USA), has yet to embrace climate change as a subject that is routinely taught. One reason may be that many teachers are not comfortable with the subject and lack the confidence to be able to teach it to their students (Lambert et al., 2012). Another survey suggests that negative emotions surrounding the topic are increasing climate anxiety among teachers, which could help explain hesitancy in climate science curriculum delivery to students (Clayton et al., 2023). When compared to other states in the USA, the topic of climate change is less prevalent in Florida's classrooms, but the state standards are not currently devoid of statements about climate education (National Center for Science Education, and Texas Freedom Network Education Fund, 2020; CPALMS, 2024). Oftentimes a standard will cover relevant lessons without specifically mentioning climate change, while other times the issue is combined with other subjects (Sabella, 2019). Without some kind of professional development, teacher support, or ability to devote time to subjects such as climate change and environmental sustainability, teachers are unlikely to approach these topics with their students. As a result, climate change has yet to make it into mainstream education in the Florida classroom.

## 1.1 Climate change education and youth empowerment in the USA

Many resources and effective strategies that teach climate change science and to some extent climate advocacy to youth-based audiences and beyond are available to teachers in the United States (Busch, 2016; Drewes et al., 2018; Filho and Hemstock, 2019; Ledley et al., 2014; Monroe et al., 2019; Rousell and Cutter-Mackenzie-Knowles, 2020; Zabel et al., 2017). In addition, youth voices and their call for climate action have been of particular note to improve accessibility to factual information that increases climate change knowledge and perhaps more importantly, provide avenues to actively and effectively participate in real solutions for society, including youth involvement in public discussions and planning processes (Busch et al., 2019; Kretser and Chandler, 2020). Organizations such as The Climate Reality Project based in Washington, D.C. (2006), The Wild Center Youth Climate Program in Tupper Lake, New York (2008) The Climate Literacy and Energy Awareness Network (CLEAN) based in Boulder, Colorado (2010), and FXB Climate Advocates based in New York, NY (2019) have all made considerable efforts to improve climate change education, information, and youth involvement within the United States. Of particular note, CLEAN developed a "rigorous iterative peer-review process" to curate digital content such as videos and visualizations that address climate and energy

literacy principals (Gold et al. 2012). Using geospatial visual tools to display complex data such as climate, weather, and currents has been an important aspect of teaching climate change (Hestness et al. 2014; Niepold et al. 2008; Wolf-Jacobs 2024).

Locally in Florida, The Climate Leadership Engagement Opportunities (CLEO) Institute based in Miami (2010) led by former science teacher and educational administrator, Caroline Lewis, provides curriculum, trainings, webinars, and more to foster climate literacy and support action-based resources. In addition, the authors of this paper from FAU Pine Jog Environmental Educational Center (FAU Pine Jog) have experience in teaching climate change and supporting teachers and youth action in Florida prior to this research. However, few Florida teachers will address the topic of climate change without administrative support or time to

prepare for those lessons unless the topic is clearly aligned with state standards.

Content standards in the United States (US) are determined by individual states. As of 2020, Florida uses the Benchmarks for Excellent Student Thinking (B.E.S.T) for math and English Language Arts (ELA), which has mixed reviews (Berner and Steiner, 2020; Friedberg et al., 2020; Wurman et al., 2020). In 2008, Florida adopted The Next Generation Sunshine State Standards for

Science (NGSSS) (Florida Department of Education, 2008). This is not to be confused with the Next Generation Science Standards (NGSS) that was adopted by at least 20 US states or used as a framework by at least another 20 US states (Bybee, 2014; Next Generation Science Standards For States, By States, 2013; OpenSciEd, Have all 50 states adopted NGSS?, 2024). All of Florida's standards can be reviewed using their portal provided by Florida State University (CPALMS, 2024). These standards are used to guide teachers through lessons and provide a source of information for Florida's K-12 Statewide Assessment Program, which

measures student success (Florida Department of Education, 2024). Therefore, teachers are expected to teach lessons with specific learning goals that target a state standard. If activities and materials do not align with a standard, then teachers are not going to be encouraged to use them. Curriculum for climate change education in Florida is written with target standards from institutions such as The CLEO Institute and FAU Pine Jog, so there seems to be more to the story.

Interestingly, a study conducted in North Carolina indicated that teachers simply did not have the time to participate in climate change professional development (Ennes et al., 2021). As a public-school teacher in Florida from 2014 to 2022, author Rachel Wellman can relate,

>    Teachers are given many tasks throughout each school day and every minute counts. We are not paid for overtime, so any prep, grading, lesson researching, and even communications with students and their parents are often outside of a
contracted paid time. Workdays can be 12 to 16 hours long for duties to be performed at high quality. Designated professional development days are often pre-planned by school districts or individual school administrators. It is difficult to balance work and homelife during a school year and most teachers need their time off to recharge, so the last thing they want to do is extra professional development.

Teacher burnout is a well-documented issue in the profession (Chang, 2009; Ghanizadeh and Jahedizadeh, 2015) and stress and

anxiety from the Coronavirus disease (COVID-19) pandemic only amplified the problem in 2020 to 2022 (Pressley and Learn, 2021). Therefore, teachers rarely seek out extra professional development beyond the requirements given by school districts and administrators in Florida.

Financial incentives can also be a factor in gaining teacher participation. Investigative reporting on the issue suggested that many

curriculum materials available on the topic of climate change in the US are generated by the energy industry. Wealthy companies will often produce and distribute curriculum materials and sponsor groups/teachers to use their materials while teaching subjects

such as climate change. These messages can confuse the cause and effects of the phenomena, increasing uncertainty, confusion, and distrust about climate science (Worth, 2021).

The good news is that we still have hope for our youth and for the future of climate change education in the US. In addition to programs already mentioned here, some success in teacher professional development has been noted at FAU Pine Jog through collaborations with Earth Force where teachers were given stipends to develop, implement, and document lesson plans that focused on student action civics related to climate change and a teacher collaborative network was formed (Environmental Action Civics, 2023; Wellman and Henderson, 2022). Students within these classrooms are using place-based and/or project-based learning to

help them be part of the solution. More success seems to come from providing services that support the students directly. Many students are motivated, dedicated, reaching out for resources and information, and are developing their own programs (Youth Climate Summits) through the support of science and education centers (Han and Ahn, 2020; Kretser and Griffin, 2020; McDonnell et al. 2011). Others are engaging in local and international meetings, designing solutions, conducting experiments, and completing petitions for local governments to take action (Climate Change Education.Org, 2024; Parker, 2020; The Sink or Swim Project,

2016). This type of learning through experience creates engaged citizens as students see their visions through successful outcomes (Burke et al., 2020). Empowering students, giving them hope for positive change in light of an otherwise dreary outlook of climate change and allowing them to investigate and provide service through action in the community will foster a stronger community for better tomorrows (Li and Monroe, 2019). The best thing we can do as mentors, parents, and teachers, is support them throughout the process.

**1.2 Program overview and theoretical framework**

This three-year program, entitled Climate Resilience Education and Action for Dedicated Youth Program (Climate READY program) was funded by the National Oceanic and Atmospheric Administration (NOAA) Environmental Literacy Program grant (NOAA-SEC-OED-2020-2006190). Using the Design and Development research model (Earle et al., 2013), the program provided the opportunity to use an original curriculum developed by FAU Pine Jog and NOAA assets to strengthen community resilience

and adaptive capacity of participants. Specifically, these students were in grades four through 12 (youth ages nine to 17 years old) with recruitment priority given to underrepresented schools and residents of the larger community in Palm Beach County, Florida. Participants learned the scientific principles behind the global and local changes in climate and studied the relationships and power dynamics of nature, self, and community, which enabled them to become agents in protecting resources and building greater resilience against extreme weather events. Participants also studied the Regional Climate Action Plan (RCAP), which is a guiding

tool for coordinated climate action in Southeast Florida to reduce greenhouse gas emissions and build climate resilience (Southeast Florida Regional Climate Change Compact, 2022). The RCAP provided a set of over 100 recommendations and guidelines for implementation and shared best practices for local entities to align with the regional agenda. The Climate READY Program was delivered over three years as described in our methods. The three main components were (1) Climate READY Ambassador Institute, (2) Climate READY After-school Program, and (3) Climate READY Community Outreach.


The design of this program was informed by the NOAA Community Resilience Education Theory of Change theoretical framework, which outlines the goals of community resilience education as follows (Bey et al., 2020):
To develop environmental literacy to understand threats and implement solutions that build resilience to extreme weather, climate change, and other environmental hazards, including the knowledge, skills, and confidence to:

1.  reason about the ways that human and natural systems interact globally and locally, including the acknowledgement of disproportionately distributed vulnerabilities;

2.  participate in civic processes; and

3.  incorporate scientific information, cultural knowledge, and diverse community values when taking action to anticipate, prepare for, respond to,

**1.3 Geographic location and hazard identification**

Implementation of all three components of the Climate READY program occurred in six specific regions within Palm Beach County in the Southeastern region of Florida. These included Boca Raton, Boynton Beach/Delray Beach, Lake Worth Beach, Riviera Beach, West Palm Beach, and the Glades areas (Pahokee/ Belle Glade) located near the Everglades Agricultural Area (Fig. 1). For South Florida, the dense population, low-lying coasts, porous geology, and distinctive hydrology characterize it as one of

the world's most vulnerable areas from the impacts of climate change. In fact, South Florida could be the next environmental "ground zero" (University of Miami Frost Institute for Data Science and Computing, 2015; see definition in Appendix 1). Sea level rise, threatening water and air quality, saltwater intrusion, and the destruction of the Everglades, are some of Florida's most pressing challenges. South Florida is exposed to nearly all of the nationally identified risks of climate change including urban infrastructure and health risks, increasing flood risks in coastal and low-lying regions, transformation of natural ecosystems, economic and health

risks for agricultural communities, increased prevalence of disease-carrying insects, increased frequency of hot weather temperatures, hurricane and storm intensification, risks associated with sea level rise and storm surge, increase in invasive species, and threatened freshwater quantity and quality (U.S. Global Change Research Program, 2016). The short - term projections for sea level rise in Southeastern Florida are between 6 to 10 inches by 2030 and 14 to 26 inches by 2060. In the longer-term, sea-level rise is projected to be between 31 to 61 inches by 2100. This rise is due to increasing ocean volume due to the thermal expansion

of water, groundwater losses, glacier mass loss, and discharge from land-based ice sheets (Heimlich, et al., 2009). According to a study by Hauer, Evans, and Mishra (2016), a sea level rise of this level would affect between 24,000 and 57,000 people in Palm Beach County alone. Additionally, according to the NOAA Theory of Change, "societal processes have created unequal exposures to environmental threats and access to solutions within a community" (Bey et al., 2020). This is certainly true in Palm Beach County. The Social Equity Key to Southeast Florida RCAP 2.0 Factsheet highlights the growing concern for low-income areas

and communities of color (Center for American Progress et al., 2018):

"Communities of color and low-income areas are disproportionately exposed to heat, flooding, and pollution risks—meaning extreme weather events often hit them hardest. In a region where city streets flood even on sunny days, and in the wake of the record-breaking 2017 hurricane season, local leaders recognize that they have little time to waste" (p.1). "Climate change threats exacerbate and multiply historic inequities that exist in low-income areas and communities

of color……. Many communities of color were purposefully sidelined by 20th-century development decisions resulting in economic and racial segregation, making it particularly difficult for communities without targeted policies and resources to build local economies that are just and resilient to climate change" (p.1). "Low-income areas and communities of color are particularly vulnerable to the effects of extreme weather because they are often located in or near flood-prone areas, heat islands—urban neighborhoods where concrete and asphalt surfaces absorb and radiate heat, producing

temperatures that are warmer than average—or toxic waste sites. They are also often overburdened by disproportionately high air and water pollution" (p.1).

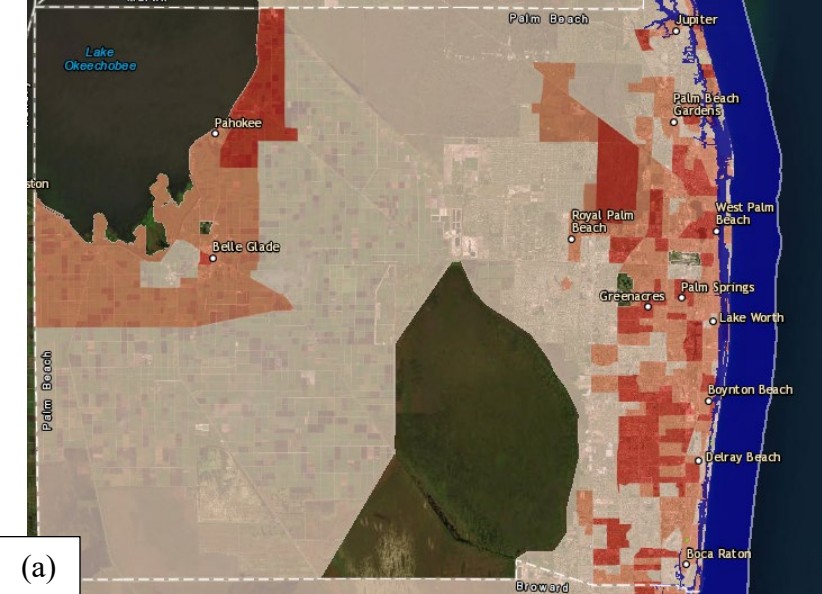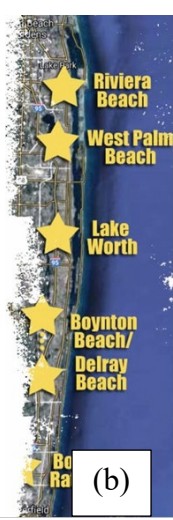

**Figure 1:** Palm Beach County Vulnerability (NOAA's Office for Coastal Management Sea Level Rise Viewer Accessed September 2023). Zones in red represent high vulnerability, personal disruption, and risk, and pink zones demonstrate medium indicators of risk and social vulnerability (a). Map of coastal communities targeted in the Climate READY Program that was used as part of FAU Pine Jog presentation template (b).

In looking at the Glades areas of Pahokee and Belle Glade (Fig. 1), as an example, the demographics reveal some of the most challenging conditions that exist in the state of Florida and most likely the nation, compounding the struggle that educators have in making climate education a priority. For example:

- 72% live in single-parent or non-traditional households;
- 74% in households earning less than $29,999 a year;
- 93% of students receive free or reduced lunch;
- 35% live in households earning less than $11,999 a year;
- Crime rates place Belle Glade as safer than only 4% of US Cities and Pahokee safer than only 10% (Belle Glade Crime Rates 2022; Pahokee Crime Rates 2022; Badcock, 2018).
- 78% are Black/African American, 5% are Hispanic, 4% are Multiracial, 12% are Undeclared, 1% are White

Other targeted areas in Florida are equally dangerous and are highly affected by poverty. In 2018, two of the target cities, Riviera Beach and Lake Worth Beach, ranked 30th and 31st as most dangerous cities in the United States (Todaro, 2018). Complicating Florida's climate and human challenges is Florida's growing population. The US Census data indicates that approximately 960 people relocate to Florida each day. Given Florida's population growth, expected to grow to 33.7 million by 2070, and development trends, new community-based approaches to conservation education and restoration are imperative (Florida Department of Agriculture and Consumer Services et al., 2017). Florida is now the third most populous state and by 2030, five million more residents will call Florida home and 1.7 million more jobs will be needed (Florida Chamber Foundation, 2017). In 2018, Florida's population of people of color under the age of 70 became a majority, at 53.5% of the population (Taylor, 2019).

**1.4 Program objectives and hypothesis**

The main goal of the Climate READY program was to increase the environmental literacy of fourth through 12th grade students in Palm Beach County, FL and the general community that they live in so that they can become more resilient to extreme weather and/or other environmental hazards, thus empowering them to become involved in achieving that resilience. Therefore, our

research hypothesis for the program was that participants in Climate READY will better understand their community strengths and vulnerabilities to a changing climate and that they will feel empowered to participate on both a personal and civic level to take action, minimize risks, adapt, and weigh the potential impacts of their decisions. This goal and hypothesis are in keeping with the NOAA Resilience Theory of Change theoretical framework, which theorizes that "environmental literacy, along with community health, civic engagement, social cohesion, and equity, enhance resilience" (Bey et al., 2020; see section 1.2). Objectives for this program are included in Table 1.

**Table 1**: Climate READY general objectives for all students, for dual enrolled high school/FAU students (Climate READY Ambassadors ages 15 to 17), and for fourth and fifth grade after-school students ages nine to 11.

| General objectives for all students |
| --- |
| 1. Increase content knowledge of the history and causes of climate change. |
| 2. Identify and evaluate personal and community strengths and vulnerabilities in response to extreme weather events. |
| 3. Acknowledge that disproportionate distribution of vulnerabilities and diverse community values exist. |
| 4. Promote environmentally responsible behavior that results in the stewardship of healthy ecosystems and a reduction in carbon consumption to mitigate future environmental risks. |
| 5. Improve critical thinking skills to assess the sources of different climate change perspectives and attitudes. |
| **Objectives specific to students (ages 15 to 17)** |
| 1. Design and implement community resilience-related service-learning projects based on local environmental challenges. |
| 2. Empower students to act as agents of change within the community by teaching community members about local climate impacts and resilience strategies for extreme weather events. |
| **Objectives specific to students (ages 9-11)** |
| 1. Design and complete a storybook on community resilience to build an understanding of climate change and the impacts facing Southeastern Florida. |

## 2 Methods

### 2.1 Program Description

The three-year Climate READY program used original lesson plans developed by FAU Pine Jog staff and NOAA assets to strengthen community resilience and the adaptive capacity of participants in six underserved regions in Palm Beach County, Florida as described in section 1.3. The Sea Level Rise Viewer developed by the NOAA Office for Coastal Management allowed us to add social and economic data overlays (Fig. 1(a)), which identified three of the target regions (Riviera Beach, Lake Worth Beach and the Glades) as red zones of high vulnerability, personal disruption, and risk (2023). The remaining regions, West Palm Beach, Boynton Beach/Delray Beach and Boca Raton were highlighted as pink zones, demonstrating medium indicators of risk and social vulnerability. While these areas have undergone gentrification in sections near the Atlantic Ocean, extreme pockets of poverty and vulnerable populations still exist in these boundaries (Sea Level Rise Viewer 2023). We targeted Title 1 schools in low socio-economic communities as defined by the Bureau of Federal Educational Programs and part of the Every Student Succeeds Act (ESSA), which identifies schools that have at least 60% of their students living in low-income households (National Center for Education Statistics, 2019; Office of Program Policy Analysis and Government Accountability, 2023).

The Climate READY program included the development of three interconnected components that were divided into three-semester courses at FAU listed here and described in the subsections below:

- **Climate READY Ambassador Institute** offered as *EDG 4930: Building Community Climate Resilience*
- **Climate READY After-school Program** offered as *EDG 4930: Youth Mentorship in Climate Resilience*
- **Climate READY Community Outreach** offered as *EDG 4930: Community Resilience Outreach*.

The new three- semester program was implemented over two full cohorts and a partial third cohort participated in a modified single semester course in summer 2023. An advisory council of local resilience experts was established during the planning phase to provide feedback about the program. Using their advice, FAU Pine Jog staff used the first year of funding from October 2020 to May 2021 to establish an original curriculum for the three-semester dual enrollment components. However, the global pandemic and temporary freeze on in-person classroom interaction required us to create an online learning approach for the first year of implementation, July 2021 to May 2022 (Cohort 1). After restrictions were lifted in 2022, the Climate READY program was implemented with face-to-face interaction as originally designed during the second year of implementation, July 2022 to May 2023 (Cohort 2). Using lessons learned from Cohort 1, we revised the curriculum for Cohort 2 and collected data as described in our Research Methodology (section 2.2).

### 2.1.1 Climate READY Ambassador (CRA) Institute

The CRA Institute was for ninth through 12th grade students, Summer 2021 and 2022; Semester 1. The summer course developed foundational climate literacy knowledge focused on current scientific research, implementation practices, and community resilience measures. Students ages 15 to 17 from underserved communities were prioritized in the recruitment process and were offered the opportunity to register as dual enrolled (high school/university) students under a special topics course in FAU's College of Education. Students that had a strong application for the Climate READY Program but did not qualify for dual enrollment at FAU were able to participate on a non-credit basis and received documented community service hours for completing the program. By design, the institute was created as an intensive five-day residential program at the FAU John D. MacArthur Campus in Jupiter, Florida, which provided the opportunity for a deeper connection and immersion into the climate literacy/resilience content, and a peer experience different from a traditional school-based setting. Field trips to Galaxy E3 Elementary School and MANG, a local mangrove nursery, were organized off campus to expose students to hands on experiences. However, due to the global pandemic and COVID concerns during the Summer 2021 semester, students in our first cohort of students were not able to participate in the full in-person residential experience. We transitioned from a residential to a hybrid model for the CRA Institute having planned the first and last Saturdays as extended in-person days at Galaxy E3 Elementary School in Boynton Beach and at FAU Pine Jog in West Palm Beach respectively. Classes Monday through Friday during that week were shorter virtual days using WebEx. With restrictions lifted in Summer 2022, student participants in the second cohort, our experimental group, were able to attend the originally designed in-person residential model (Fig. 2).

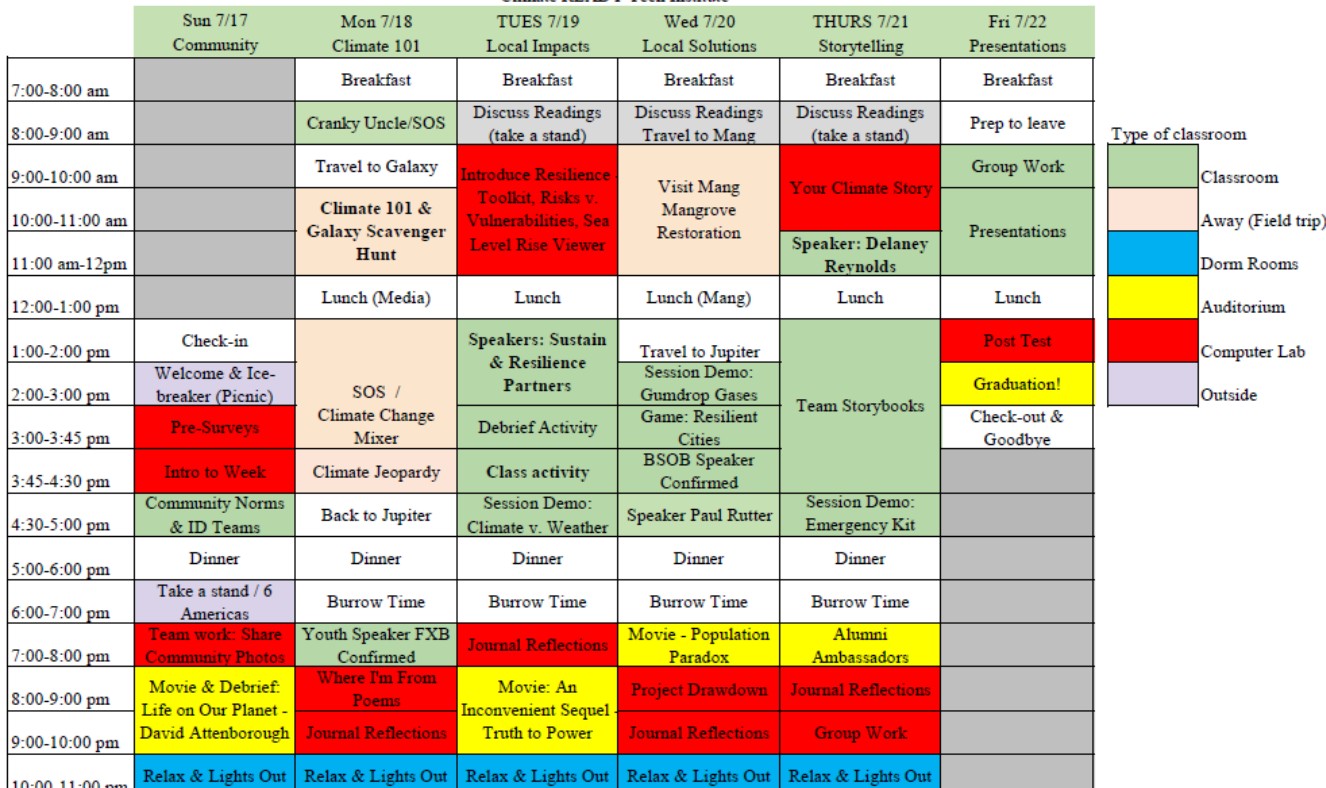

| | Sun 7/17 Community | Mon 7/18 Climate 101 | TUES 7/19 Local Impacts | Wed 7/20 Local Solutions | THURS 7/21 Storytelling | Fri 7/22 Presentations |
|---|---|---|---|---|---|---|
| 7:00-8:00 am | | Breakfast | Breakfast | Breakfast | Breakfast | Breakfast |
| 8:00-9:00 am | | Cranky Uncle/SOS | Discuss Readings (take a stand) | Discuss Readings Travel to Mang | Discuss Readings (take a stand) | Prep to leave |
| 9:00-10:00 am | | Travel to Galaxy | Introduce Resilience Toolkit, Risks v. Vulnerabilities, Sea Level Rise Viewer | Visit Mang Mangrove Restoration | Your Climate Story | Group Work |
| 10:00-11:00 am | | Climate 101 & Galaxy Scavenger Hunt | | | Speaker: Delaney Reynolds | Presentations |
| 11:00 am-12pm | | | | | | |
| 12:00-1:00 pm | | Lunch (Media) | Lunch | Lunch (Mang) | Lunch | Lunch |
| 1:00-2:00 pm | Check-in | | Speakers: Sustain & Resilience Partners | Travel to Jupiter | | Post Test |
| 2:00-3:00 pm | Welcome & Ice-breaker (Picnic) | SOS / Climate Change Mixer | | Session Demo: Gumdrop Gases | Team Storybooks | Graduation! |
| 3:00-3:45 pm | Pre-Surveys | | Debrief Activity | Game: Resilient Cities | | Check-out & Goodbye |
| 3:45-4:30 pm | Intro to Week | Climate Jeopardy | Class activity | BSOB Speaker Confirmed | | |
| 4:30-5:00 pm | Community Norms & ID Teams | Back to Jupiter | Session Demo: Climate v. Weather | Speaker Paul Rutter | Session Demo: Emergency Kit | |
| 5:00-6:00 pm | Dinner | Dinner | Dinner | Dinner | Dinner | |
| 6:00-7:00 pm | Take a stand / 6 Americas | Burrow Time | Burrow Time | Burrow Time | Burrow Time | |
| 7:00-8:00 pm | Team work: Share Community Photos | Youth Speaker FXB Confirmed | Journal Reflections | Movie - Population Paradox | Alumni Ambassadors | |
| 8:00-9:00 pm | Movie & Debrief: Life on Our Planet - David Attenborough | Where I'm From Poems | Movie: An Inconvenient Sequel - Truth to Power | Project Drawdown | Journal Reflections | |
| 9:00-10:00 pm | | Journal Reflections | | Journal Reflections | Group Work | |
| 10:00-11:00 pm | Relax & Lights Out | Relax & Lights Out | Relax & Lights Out | Relax & Lights Out | Relax & Lights Out | |

**Type of classroom**

| | |
|---|---|
| 🟩 | Classroom |
| 🟧 | Away (Field trip) |
| 🟦 | Dorm Rooms |
| 🟨 | Auditorium |
| 🟥 | Computer Lab |
| 🟪 | Outside |

**Figure 2:** Schedule of activities and lessons implemented during Semester 1 of the Climate READY Program (July 2022), a weeklong residential college course designed for dual enrolled high school students, the Climate READY Teen Institute. For the exception of two field trips to Galaxy E3 Elementary School and MANG Nursery (tan), classrooms (green) were within walking distance to dorms (blue) and dining hall (white) where vegan and vegetarian options were given during all meals.

Climate science was taught in part by using NOAA's Science on a Sphere (SOS) technology in collaboration with Galaxy E3 Elementary School (Galaxy E3), a Title-1 public school (see definition in Appendix 1) with Platinum level LEED (Leadership in Energy and Environmental Design; see definition in Appendix 1) Certification (Kubba 2008) serving a primarily low socio-economic population in Boynton Beach, Florida vulnerable to climate change (Fig. 1). Galaxy E3 is a unique school that received generous community donations to rebuild after their old school was condemned and demolished (Storyology Studios Presents: Galaxy E3 Elementary School, 2015). The majority of the students in regular attendance at Galaxy E3 are ages five to 11 and live at or below the poverty line (U.S. Department of Health and Human Services – Poverty Guidelines, 2024; and see definition in Appendix 1). The school was specifically chosen to be a partner in the FAU Climate READY Program due to its success as a top tier rated (Platinum) LEED certified facility that implemented environmentally sustainable methods during construction while also featuring nontraditional teaching tools such as NOAA's SOS technology (U.S. Green Building Council Inc. – LEED rating system, 2024). In addition, FAU Pine Jog program coordinators and staff have worked with teachers and staff at Galaxy E3 over the years on numerous occasions and professional connections were in place prior to this project's inception. NOAA's SOS technology was used to help facilitate our Climate 101 lessons. The SOS uniquely projects seamless imagery on a sphere-shaped projection screen that is six feet in diameter and suspended from the ceiling of a dedicated room. It has shown to effectively enhance student learning of weather and climate concepts using global scale earth system science (Rowley et al. 2013), which is in line with the concept of using geospatial visualizations to enhance climate science literacy (Hestness et al. 2014; Niepold et al. 2008; Wolf-Jacobs 2024). We created a SOS playlist for the students using data provided by the NOAA SOS datasets (Science On a Sphere Dataset Catalog

2023; See Appendix 2, Table 5). This playlist was an important part of teaching climate change science as it provided visual representations of global carbon dioxide concentrations, global temperature, global hurricane pathways, and changes in arctic sea ice coverage over time. A few of the video shorts created by NOAA partners that were included in the NOAA SOS datasets were also shared. In using the SOS, students were able to connect what they learned about the causes and effects of climate change with visual global patterns through time, which reinforced concepts such as the difference between climate and weather, the patterns of greenhouse gas emissions and temperature changes, and how communities can work together towards a common goal. Teen participants in the CRA Institute would later use the SOS with younger (after-school) students in the fall semester.

Throughout the weeklong summer course, CRA Institute students connected with sustainability officers, scientists, researchers, and representatives from local government and the RCAP community. Using place-based active learning strategies, this institute focused on anthropogenic issues impacting South Florida, such as sea-level rise and extreme weather events. Lessons delivered included using a case-study approach, where participants evaluated the risks, assets, and vulnerabilities of their local municipality and explored inequities produced by current systems. Major assignments for students during the summer component included a photovoice project to help students connect with their communities (Photovoice, 2023; Science, Camera, Action! 2023), a "Where I'm From" poem and puzzle piece class project (Christensen, 2001; see Appendix 2, Fig. 10), the creation of their own "Climate Story" (Discover Your Climate Story, 2006; example in Fig. 9(b)), and a team assignment to create a storybook using an original template created by FAU Pine Jog (see Appendix 2, Fig. 12). The storybook lesson was important to help connect with the younger students in the after-school component during Semester 2. In both cohorts, students that completed the 80-hour program were equipped as Climate READY Ambassadors (CRAs) that were responsible for delivering the second component, the Climate READY After-school Program.

### 2.1.2 Climate READY After-school Program – Youth Mentorship

The Climate READY After-school Program was designed for the ninth to 12th grade students (ages 15-17) that completed the CRA Institute the previous summer, naming them CRAs; Semester 2. These students worked as mentors to fourth through fifth grade after-school students (ages nine to 11) while enrolled in FAU College of Education's *Youth Mentorship in Climate Resilience* in the Fall semester immediately following the summer semester for each cohort. Students were given the option to take the course as a dual enrolled student or for community service hours as a non-credit course. FAU Pine Jog staff met with the class during five, four-hour online classes using WebEx or Google Meets on Saturday mornings throughout the semester, and a final all-day in-person class to conclude the course and participate in a field experience. The CRAs were also required to meet with their after-school groups. This course was created as the second part of the three-semester CR Program to connect with local elementary after-school programs where students delivered grade level appropriate climate resilience activities using four detailed 60-minute sessions aligning with Florida State Standards (CPALMS, 2024); *Session 1 - Climate Basics, Session 2 - Local Solutions, Session 3 - Storytime!, and Session 4 – SOS Adventure!* (Table 2).

**Table 2**: Climate READY After-school lessons and activities implemented by the Cohort 2 CR Ambassadors during the fall semester (2022).

| After-school Session | Lesson | Guiding Questions and Topics Covered | Activities |
|---|---|---|---|
| 1 | Climate Basics | 1. What is the difference between weather and climate?<br>2. Why is the climate changing?<br>3. What is the greenhouse effect?<br>4. What are some greenhouse gases (& where do they come from)? | 1. After-school Pre-Survey<br>2. Climate vs. Weather Activity<br>3. Video Viewing Guide |
| 2 | Local Solutions | 1. What are some ways climate change could impact us locally?<br>  a. Extreme Heat<br>  b. Rising Seas<br>  c. Stronger Storms<br>2. What does community resilience mean?<br>3. What are some things we can do to be more resilient/deal with those impacts? | 1. Introduce CR Ambassador Team created Climate Resilience Storybook<br>2. Storybook activity using FAU Pine Jog Community Resilience Storybook template. After-school students chose:<br>  a. Main Character<br>  b. Climate Impact<br>  c. Action/Solution<br>  d. Community Helpers |
| 3 | Storytime! | 1. Review of story elements chosen by after-school students:<br>  a. Main character<br>  b. Impact<br>  c. Solution<br>  d. Community helpers<br>2. CR Ambassadors deliver original story created from the after-school student choices using the FAU Pine Jog Community Resilience Storybook template. | 1. After-school students illustrate the new community climate resilience story using crayons, markers, and their imaginations (See Appendix).<br>  a. Example title and cover:<br> |
| 4 | SOS Adventure! | 1. Field trip from after-school site to Galaxy E3 Elementary School<br>2. Introduction to NOAA's Science on a Sphere<br>3. Sustainable features of a LEED Platinum certified school<br>4. Extreme weather preparedness | 1. Presentation/reading of completed class storybook<br>2. Introduction to NOAA's Science On a Sphere<br>3. Tour of Galaxy E3 Elementary Energy-Saving Features<br>4. "Get Storm Ready" game<br>5. Program Post-Survey |

340    Guided by educational professionals from FAU Pine Jog, CRAs worked in teams of four to five, to deliver over four hours of programming to up to six after-school sites in Palm Beach County during each program cycle. After-school programs that targeted underserved communities were given priority in the recruitment process. Activities focused on gaining a conceptual understanding of basic climate processes, potential impacts and local resilience initiatives that are already in place and empowering all students through action by creating a storybook focused on community specific needs (Table 2). The CRAs used an original template

345    created by FAU Pine Jog to assist the fourth and fifth grade after-school students ages nine to 11 in developing an illustrated storybook of their own, which highlighted how their community could be resilient to climate change (see Appendix 2, Fig. 12). The original storybooks were printed and bound through an online company, DiggyPOD (2004; see Appendix 2, Table 6), and distributed to the after-school students and CRAs that participated in the program to take home and share with their families. Any

extra books were given to the participating school to share in their school or classroom libraries. The culminating Lesson 4 provided transportation for all students to visit Galaxy E3 Elementary School in Boynton Beach, Florida to learn about how the school achieved Platinum level LEED certification and for the students to experience standards-based content on NOAA's SOS technology. Lastly, a FAU Pine Jog original game called "Get Storm READY" was used to help teach students the importance of preparing for a hurricane and gave them time to think about what items may be important when creating a hurricane preparedness kit for their families (Fig. 8(c)).

### 2.1.3 Climate READY Community Outreach

The final semester of the Climate READY Program was for the community at large, delivered by CRAs in Spring 2022 and 2023 – Semester 3. It included the FAU College of Education's course *Community Resilience Outreach*. Similar to previous semesters. The CRAs were given the choice to take the course for dual enrollment credit or for non-credit and receive community service hours. FAU Pine Jog staff met with the class during five, four-hour online classes using Google Meets on Saturday mornings throughout the semester, and a final all-day in-person class to conclude the Climate READY Program and participate in a field experience. This component of the program emphasized building knowledge and skills to implement climate resilience education curriculum and activities within local communities (Table 1). The design of this course was informed by the NOAA Community Resilience Education Theory of Change, which outlines the goals of community resilience education (Bey et al., 2020; see sections 1.2).

Lessons also addressed social emotional learning as well as time management practice to help better prepare our youth for public speaking and to help them with time commitments (see Appendix 2, Fig. 11). The Climate READY Ambassador teams were required to create a Community Resilience Plan tailored to their assigned communities. The CRAs researched their assigned community's assets, strengths, and vulnerabilities, and then recommended at least three possible solutions to help solve the issues their communities are facing with the threat of climate change. Each team was then required to present their resilience plan by participating in two presentations to members of their respective communities. One presentation needed to be a formal lecture and discussion style presentation using a PowerPoint or Google Slides format while the second presentation could be a table event at a festival or fair. Each presentation focused on a Community Resilience Plan that included CRA research on place-based needs to climate resilience throughout the three-semester program. In their presentations, CRAs outlined how their communities could be more resilient to the impacts of climate change, the strengths of their respective communities, and potential solutions to help mitigate and/or adapt to impacts. FAU Pine Jog instructors coordinated the outreach events for Cohort 2 in the spring semester (see Appendix 2, Fig. 11). Graded "Time Management" lessons were designed and implemented during class time for the dual enrolled students to teach them the importance of scheduling and keeping commitments.

Four of the 13 graded outreach events during Cohort 2 were not on this student schedule. These events were planned and implemented prior to this Time Management assignment (see Appendix 2, Fig. 11). Events included three Urban Sustainability Directors Network (USDN) grant collaborations with the Palm Beach County Office of Resilience where students interacted with members of the community in Pahokee, Belle Glade, and West Palm Beach (Amplifying Impact Community Partners of South Florida, 2023; Fig. 9(a-c)). The fourth event, Climate and Art, was coordinated by the Delray Beach Office of Sustainability and Resilience. Students in the Boynton Beach/Delray Beach Team shared the storybook they created during the summer CRA Institute with the Delray community (Climate and Art Delray Beach 2022).

**2.2 Research Methodology**

Using lessons learned from Cohort 1, we revised the curriculum for Cohort 2 and collected data using a pre-questionnaire in the beginning of the Climate READY Institute (July 2022), a post-questionnaire after the Climate READY Institute in July 2022 (Post 1), and we repeated the post-questionnaire at the end of the last semester course that focused on community outreach in May 2023 (Post 2). The three data points were used to assess learning outcomes, retention of content throughout the three-semester program, and to test our hypothesis in this study.

The CRAs were also given pre- and post- questionnaires where they were asked to describe the degree of their connection with their communities using circles to represent sense of place within a community (Fig. 3) and to evaluate self-efficacy. The circle labelled "Y" represents the student and the circle labelled "C" represents their community. These circles were shown in four different settings representing the degree of closeness between self and community. The CRAs were asked to indicate which setting best represented them followed by space for the student to explain why they chose that setting.

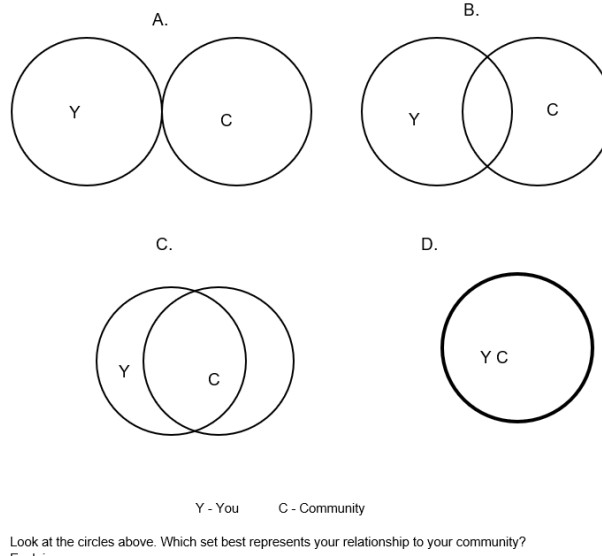

**Figure 3:** Survey question using circles to represent sense of place within a community. The circle labeled "Y" represents the student and the circle labeled "C" represents their community. Adapted from Petersen et al. (2020).

**2.2.1 Curriculum evaluation**

The evaluation utilized a mixed methods approach focused on measuring changes to participant Behaviors, Attitudes, Skills, Interests, and Knowledge (BASIK) (Creswell, 2006; Davis et al., 2019; Frechtling, 2010). Participants within this study included the Corhort 2 CRAs that completed the three-semester dual enrollment program with FAU Pine Jog during the July 2022 to May 2023 academic year. Summary reports were filed after each phase of the program as well as an annual report presenting both formative and summative findings and offering recommendations to the NOAA grant co-principle investigators to FAU Pine Jog Program Coordinators and supporting staff.

**2.2.2 Pre- and post-assessment questionnaires and surveys**

A questionnaire was created by the writers of this paper with 32 content knowledge questions to assess CRA awareness of climate change science. Many of the questions came from the "Climate Literacy Quiz" published by the Climate Literacy and Energy Awareness Network (Climate Literacy Quiz, 2018). Additional resources for content knowledge questions included texts used in

high school and college level environmental science and management courses (Butz, 2008; Myers and Spoolman, 2014; Friedland and Relyea, 2015). Using the "Global Warming Six Americas" as a gauge, CRAs were asked to share their perception of climate change ranging from dismissive to alarmed (Maibach et al., 2011). Pre- and post- surveys were also used to ask CRAs to describe their connection with their communities using circles to represent sense of place within a community (Fig. 3) and to evaluate self-efficacy. We collected information on the demographics of student participants such as age, gender, race, and grade level.

A modified pre- and post- questionnaire was also created for the after-school students; one pre-questionnaire was given before Lesson 1 in the After-school Mentorship component and an identical post-questionnaire was given at the end of Lesson 4. This survey contained fewer questions with simpler language to target after-school student learners. It was given anonymously, so no identifiers of name or location were asked.

### 2.2.3 Data Analysis

Climate READY Ambassador (CRA) -responses to pre- and post-assessment questionnaires were categorized into appropriate groups based on the individual response to each question, and then responses summarized (N=22 matched pairs). The CRA numeric responses were summarized and appropriate matched pair statistical analysis (Student's t-test) was applied. The CRAs scored responses to pre- and post-assessment questions were then analyzed for comparison using a two-sample t-test when data followed the normal probability distribution and the non-parametric Kolmogorov–Smirnov (KS) test when the probability distributions were non-normal. Open response items were analyzed with Kernal Analysis and summarized. The significance level (alpha value) was set to 0.05, and results were considered statistically significant if $p<0.05$.

In a separate analysis, the after-school student responses were anonymous and the mean values for pre- and post- questionnaire followed the normal probability distribution. Mean values for pre (N=60) and post (N=52) questionnaire were analyzed for differences using a Student's t-test ($p<0.05$).

### 3. Results overview

Data were collected and analyzed using the methods as described in Sect. 2 for Cohort 2 (July 2022 to May 2023). Although FAU Pine Jog staff spent time recruiting students in the Pahokee/Belle Glade area for Cohort 2, no student from that area completed the application process. This contrasted with Cohort 1 where six students from Pahokee/Belle Glade were active CRAs. Therefore, for Cohort 2 we focused on the five remaining communities of Boca Raton, Boynton Beach/Delray Beach, Lake Worth Beach, West Palm Beach, and Riviera Beach (see Fig. 1(b)). A total of 22 matched pair responses were summarized for the Pre and Post 1 data. Twenty responses were summarized for the Post 2 data. Most CRAs (96%) were between 15 and 16 years of age and female (72%), 55% of the CRAs described themselves as White, 36% as Hispanic or Latino, and 18% as Black. The CRAs were going to be in 10th (32%), 11th (41%), and 12th grades (27%) and they reported learning about climate change from school (3.0/5), the internet and social media (3.1/5) and television (2.7/5) primarily.

Upon the completion of the three-semester course experience with FAU Pine Jog, CRAs felt significantly more connected with their communities, and more confident in communicating their knowledge to public groups (Table 3). After their experience, they were statistically more likely to feel "Alarmed" about climate change (Table 4) and significantly fewer CRAs reported that they do not question climate change (see Appendix 3, Table 9). The CRAs showed significant improvements on 23 of 32 content knowledge items from pre to Post 1 with high pre- scores on seven items leaving no room for significant improvement (see

Appendix 2, Tables 10-22). When asked to identify some of the impacts of climate change affecting South Florida, post experience, CRA responses were more detailed, factual, and considered multiple areas of impact (see Appendix 3, Tables 21 and 22).

### 3.1 Community impact and student perception of climate change

Over 700 community members were impacted during Cohort 2 (Fig. 4). Twenty-five high school students were recruited for the dual enrollment summer session Climate READY Teen Ambassador Institute to become Climate READY Ambassadors (CRAs). Twenty-four of the CRAs returned for the fall semester after-school mentorship course where CRAs led five after-school programs located in underserved communities and designated as Title 1 (Office of Program Policy Analysis and Government Accountability, 2021). Eighty-one after-school students ages nine to 11 participated in the CRA led climate change resilience lessons and produced

five community resilience stories. These stories were made into printed books and given to all student authors (CRAs and after-school students) and school administrators for classroom distribution as described in in our methods (see Appendix 3, Table 6). Twenty-three CRAs returned for the final spring semester course and led a total of 13 community outreach engagement events, impacting over 600 individuals and families within their communities.

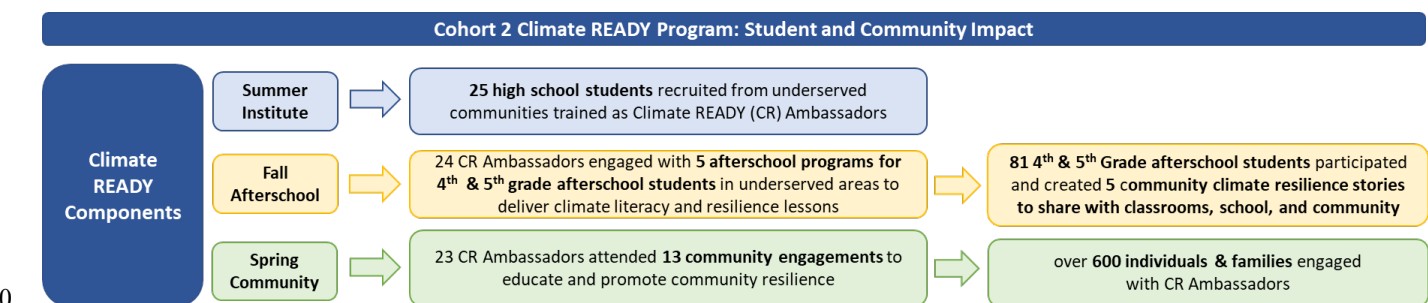


**Figure 4:** Over 700 community members including high school and elementary school students were impacted during cohort 2 of the Climate READY Program between July 2021 and May 2022.

When CRAs were asked to describe their connection with their communities using circles to represent sense of place within a

community (Fig. 3) they detected a significant ($p<0.05$) increase between pre- and post- survey responses indicating that CRAs felt closer to their communities after the three-semester CR program. Space was given for a written explanation (Fig. 3) and one CRA stated "I believe C best represents my relationship with my community, which has changed as it was originally B. I feel a lot more involved with my community after all the civic engagement projects I did with Climate READY and at school." Another CRA reported "I believe that C best represents my relationship to my community since I did many events that built the relationship.

For example, the climate presentation I did over at the Lantana Road Library allowed me to connect to the fellow residents of Lake Worth as I got to share them my experience with climate change." The CRAs also reported feeling their community was very special to them, that they like to visit places in their community, and that their community means a lot to them. By the end of the three-semester program more CRAs felt confident in helping their communities be more resilient to climate change (Table 3).

**Table 3**: Climate READY Ambassador students (CRAs) felt more confident in communicating climate change and resilience in front of a public group by the end of the three-semester program.

When you think about helping your community be more resilient to climate change, how well do you think you would be able to do the following? Rank each of the following statements both BEFORE and AFTER your experience using the following scale: 1-Defintely Can't, 2-Probably Can't, 3-Maybe Can, 4-Probably Can, 5- Definitely Can

|  | Pre | Post 1 | Post 2 |
|---|---|---|---|
| Create a plan to address the issue of climate change | 3.6/5 | 4.3 | **4.7*** |
| Get other people to care about this issue | 3.9 | 4.1 | **4.8*** |
| Organize and run a meeting about this issue | 3.7 | 4.6 | 4.5 |
| Express your views about climate resilience in front of a group of people | 4.4 | 4.4 | **4.9*** |
| Identify individuals or groups who could help me with this issue | 3.7 | 4.6 | 4.6 |
| Write an opinion letter to a local newspaper about this issue | 4.0 | 4.6 | 4.5 |
| Call someone on the phone that I had never met before to get their help with this issue | 3.1 | 3.7 | **4.3*** |
| Contact an elected official about this issue | 4.0 | 4.1 | 4.2 |
| Organize a petition about this issue | 3.7 | 4.4 | 4.0 |

*Indicates a significant difference pre to post (p<0.05)

Twenty-two of the 23 CRAs in Cohort 2 completed all three pre- and post-assessment questionnaires over the course of the three-semester program. To help gauge student perception of climate change, CRAs were asked "Overall, what are your feelings about climate change?" and the "Global Warming's Six Americas" were given as answer options to choose from (Maibach et al., 2011).
Significantly more CRAs (p<0.05) felt alarmed after the weeklong CRA Institute in Summer 2023 and significantly less CRAs (p<0.05) felt alarmed after the completion of the three-semester program (Table 4).

**Table 4**: Significantly more Climate READY Ambassador students felt alarmed about climate change from pre to post 1 assessment, with a 7% decrease in feeling alarmed by the end of the three-semester program.

**Data Summary**

Overall, what are your feelings about climate change?

|  | Pre | Post 1 | Post 2 |
|---|---|---|---|
| *Alarmed* –convinced it's happening, human-caused, a serious and urgent threat, already taking personal action and support aggressive national action. | 59% | **82%*** | **75%*** |
| *Concerned* – convinced that it is a serious problem and support a national response, not taking very much personal action. | 36% | 18% | 20% |
| *Cautious* – believe that it is a problem, but less certain that it is happening, and do not feel a sense of urgency to deal with the issue. | 5% | 0 | 5% |
| *Disengaged* –have not thought much about the issue at all and do not know much about it. | 0 | 0 | 0 |
| *Doubtful* – do not know whether to believe it is happening or not, think it may be caused by natural changes and will not harm humans for decades if at all. | 0 | 0 | 0 |
| *Dismissive* – believe that it is not occurring and is not a threat to either humans or nature. | 0 | 0 | 0 |

*Indicates a significant difference pre to post (p<0.05)

When asked "Do you think your community is already being affected by climate change?" CRAs reported a significantly greater
perception of "A great deal" throughout the course of the program. In addition, they thought that a lack of science knowledge on the topic was the main reason that people question climate change and that after completing the three-semester program 85% of the CRAs said they "do NOT question the science of climate change" (see Appendix 3, Table 9).

**3.2 Climate READY Ambassador Student climate change content knowledge Pre, Post 1, and Post 2**

The CRAs showed improvement in their understanding of climate change science throughout the three-semester program (see Appendix 3, Tables 10-22). When asked "if the greenhouse effect is natural, then why is today's climate change a bad thing?
(Select all that apply)," significantly more ($p < 0.05$) CRAs responded with correct answers during the Post 1 questionnaire (see Appendix 3, Table 10). Interestingly, 100% of CRAs responded correctly in 4 items of the same question during the Post 2 questionnaire, though it was not a statistically significant difference (see Appendix 3, Table 10). More CRAs understood that a major concern about climate change is that if $CO_2$ levels in the atmosphere exceed 450ppm, the effects would be irreversible (see Appendix 3, Table 11). The CRAs also understood that our best approaches to climate change are adapting and mitigating (see Appendix 3, Table 12), that restoring natural forests would be the best carbon sequestration strategy with the highest probability of success (see Appendix 3, Table 13), and that we need to stop burning fossil fuel before 2040 as research indicates (see Appendix 3, Table 14) (Climate Literacy Quiz, 2018; Maniatis et al., 2021).

The CRA responses to six content knowledge items on the Post 2 questionnaire were significantly different from those on Post 1 ($p < 0.05$) (see Appendix 3, Tables 15-20). These changes in responses suggest that the CRAs continued to develop their climate science knowledge throughout the year and their engagement with the Climate READY program. The CRA answers were more detailed and factual during Post 1 and more concise in Post 2 with the top three answers being sea levels rising, extreme heat, and stronger storms/hurricanes in both (see Appendix 3, Table 21 for ranked tallies of CRA responses and Table 22 for detailed CRA responses). They also showed a more scientific understanding of climate change in the South Florida area (see Appendix 3, Tables 21-22).

**3.3 After-school student pre- and post- questionnaire results.**

A total of 60 after-school student learners ages nine to 11 completed the pre- questionnaire before the Climate READY Ambassador led Lesson 1 and 53 completed the post- questionnaire after Lesson 4 (see methods). Significantly fewer after-school students reported being "not at all" worried about climate change after their experience. Significantly more after-school students felt that people were causing climate change after their experience with fewer feeling it was part of a "natural cycle" (see Appendix Table 23). Significantly more after-school students correctly identified the greenhouse effect as being "gasses in the atmosphere that trap heat" after their experience (see Appendix 3, Table 25). Significantly more after-school students understood that extreme heat and rising seas are ways that climate change could affect South Florida after their experience (see Appendix 3, Table 26). After-school students were significantly more able to identify ways of to help reduce climate change including walking or riding a bike in place of driving, unplugging TVs and computer when not in use, turning off lights when leaving a room, and restoring coastal habitats (see Appendix 3, Table 27). No significant changes were found in how the after-school students felt about being able to solve problems in their community, knowing how to make their community a better place, feeling like they can make a difference in their community, and feeling like they can make a difference in protecting the environment.

**4. Discussion**

The three-semester model for teaching and training high school dual enrolled students at FAU Pine Jog to be Climate READY Ambassadors has proven to be an effective way to empower our youth and to engage the community in climate resilience education and action. In its second year of implementation, the Climate READY program was able to provide a pathway for 23 Climate

READY Ambassadors ages 15 to 17 to participate in the global youth climate movement (Cloughton 2021) and reach as many as 700 local South Florida community members of all ages.


### 4.1.1 Objectives for all students

The program focus was to build environmental literacy of fourth through 12[th] grade students ages nine to 17, teachers and the public so they are knowledgeable of South Florida's changing climate systems, the ways in which their community can become more resilient to extreme weather and/or other environmental hazards, and how they can become involved in achieving that

resilience (Table 1). The Climate READY Program accomplished this by creating and implementing a comprehensive three-semester dual enrollment opportunity for high school students in Palm Beach County (the CRAs) as described in our methods. The CRA coursework and feedback through pre- and post- survey questionnaires demonstrated an increase in CRA understanding of climate change (see Appendix 3, Tables 7-20; $p<0.05$) and CRA responses to identifying impacts on Florida were more detailed, factual, and considered multiple areas of impact (see Appendix 3, Tables 21-22). The most significant changes came as CRAs

gained more experience as Climate READY Ambassadors (CRAs). From the summer to the spring, they became more aware of the effects of climate change on their communities, shifted their thinking about the reasons people question climate change to be more focused on the science and less on politics or religion, and increased their knowledge about key concepts associated with climate change and climate resilience.

### 4.1.2 Objectives - Summer 2022 Climate READY Ambassador Institute

The first semester (Summer 2022) worked through objectives (Table 1) that increased student content knowledge of the science of climate change (Fig. 2), improved critical thinking skills to assess the sources of different climate change perspectives and attitudes, highlighted local impacts within South Florida (see Appendix 3, Tables 21-22), explored possible local solutions (i.e. mangrove restoration with MANG, Fig. 2, Fig. 5), met professionals that work towards equitable solutions such as those within the Southeast

Florida Climate Change Compact (2023, Fig. 2), and provided a starting point for students to communicate this knowledge through creating storybooks (See Appendix 2, Table 6).

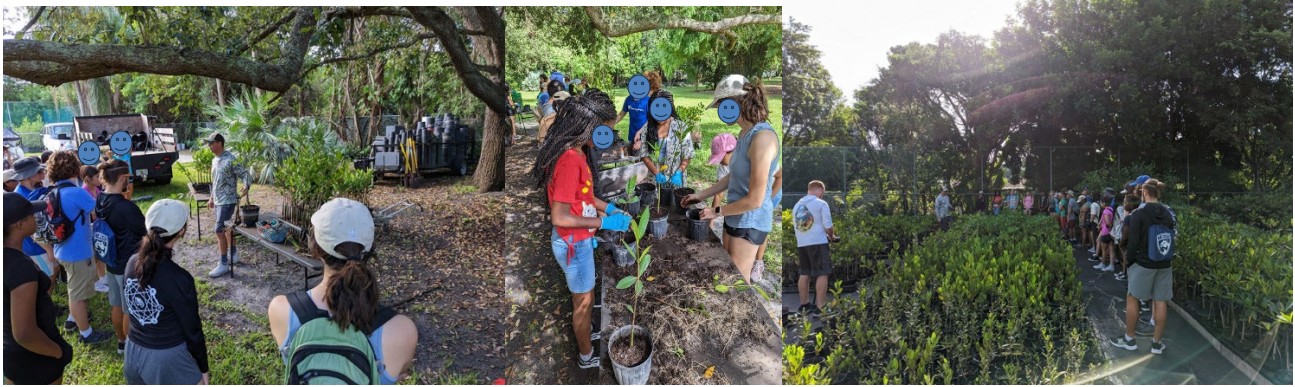

**Figure 5:** Climate READY Ambassador students participating in a service-learning field activity with MANG nursery in West
Palm Beach, FL in the Summer 2022. Co-founder Keith Rossin taught students how mangroves are used in coastal restoration and guided them through methods to grow and plant them.

In addition, the Climate READY Program successfully established relationships between CRAs across Palm Beach County through the in-person residential summer semester course. On the last day of the class CRAs were asked in an informal setting, "Think
about all the things we have done this week…the speakers, assignments, etc. What word or short phrase comes to mind?" using a

word cloud generator (Mentimeter, 2023) where words become larger when used multiple times. The final image indicated that the largest and most used words were "fun, resilience, climate change, and science on a sphere" followed by "community, adaptation, and mitigation" (Fig. 6). Many other words were added to create a large word cloud that represented the wealth of information they received from the course and most importantly, reinforcing the evidence in the survey results that the information 605 was retained. This was all while CRAs reported to have "fun," indicating that they not only learned through the experience, but that it was also enjoyable. The use of fun activities in the learning environment is often used with younger students, though previous research indicates that it can be used with all age groups and that experiencing fun and enjoyment can be "identified as a proven way to build a socially connected learning environment," (Lucardie, 2014), which can make a lasting impression on a learner and help them retain information. This is also seen in storytelling lessons (Miley, 2009).

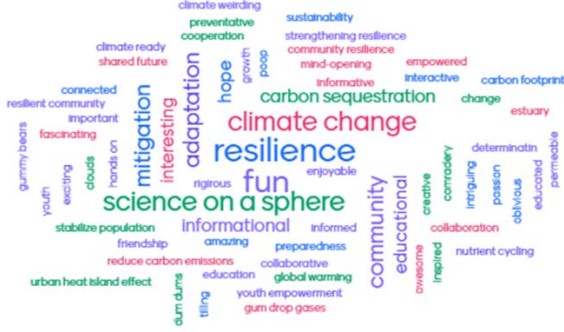


**Figure 6:** Word cloud (Mentimeter, 2023) used to generate a visual representation of student perception of the summer semester 2022, the first of the 3 semesters in the FAU Climate READY Program.

Storytelling has been used in communicating scientific concepts including climate change (Coren 2022; Moezzi et al. 2017) and 615 research has shown that storybooks can be used to bring communities together (Peters, 2023). For example, one way to teach students how to communicate complex concepts such as climate change is by creating and sharing storybooks with elementary school students. This idea came from a four-year collaboration with students from Boca Raton Community High School in Boca Raton, FL and Galaxy E3 Elementary School in Boynton Beach, FL within the Palm Beach County School District, a public school system (Fig. 7) while mentored by retired environmental science teacher, Barbara Riley. In 2018 high school students were given 620 an assignment to create a storybook of an ecological issue as part of an Advanced International Certificate of Education (AICE) Environmental Management course (Cambridge International AS Level Environmental Management – (8291), 2022). The purpose of the lesson was for high school students ages 15-18 to research an ecological issue of choice and to identify the main causes, effects, and possible solutions. The exercise required these students to breakdown complex issues into smaller digestible units that could be easily described to an elementary school student (ages 5 to 11). At the same time, creative freedom was given, which 625 gave the high school students an opportunity to gain confidence in communicating the subject. As part of the course, these high school students took a field trip to share their stories with kindergarten through second grade students at Galaxy E3 Elementary School where the high school students became the teachers. While at Galaxy, students were also exposed to another form of storytelling through NOAA's SOS technology to reinforce earth system science concepts they were teaching earlier that day. Due

to a successful first year of implementation, the collaboration continued for an additional three years, until Spring 2022, with a virtual year in 2020 during the global pandemic (Fig. 7; NOAA Grant Helps Teach Students About the Environment, 2021).

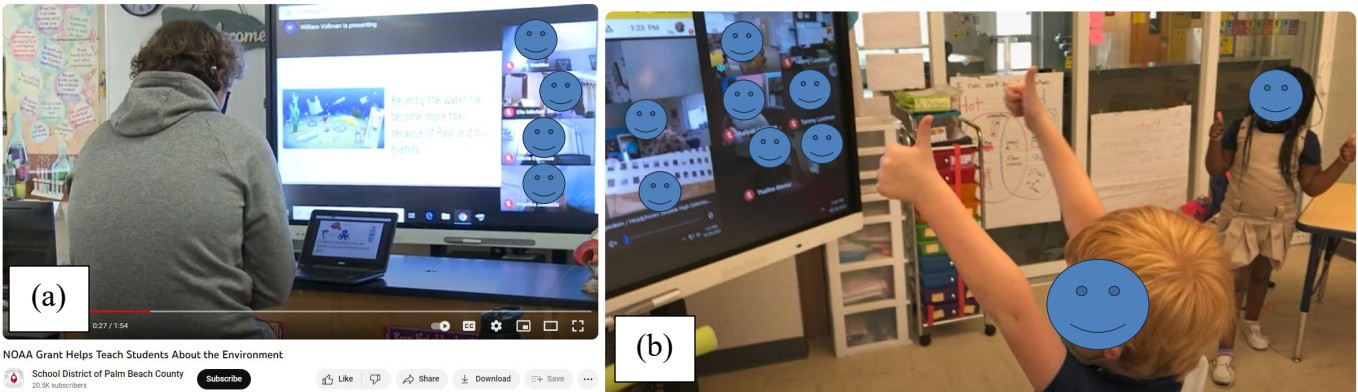

**Figure 7:** AICE Environmental Management student from Boca Raton Community High School (a) sharing his ecological storybook project with kindergartners at Galaxy E3 Elementary School (b) (NOAA Grant Helps Teach Students About the Environment, 2021).

Classroom surveys were conducted to analyze student feedback and although the results of the four-year program are not currently published, the testimonials of both elementary and high school students were noted and used to improve the lesson for the AICE course. One student reported "Whatever you do, keep the storybook activity and field trip to Galaxy. I will never forget my experience with the little kids" after being asked for feedback about the overall course. The storybook lesson impacted everyone involved including the students, teachers, volunteers, and administrators. One teacher reported "This experience gave me goosebumps!" and "I even learned something new today." The storybook lesson brought the community together and it gave them a safe space to talk to each other about environmental issues and solutions. Training teenagers (15 to 18 years old in this case) as teachers can have lasting effects on student confidence and character growth (Worker et al., 2019). The Climate READY Program took this idea a step further by creating a unique storybook template that highlights climate change in the local community (see Appendix 3, Table 6; Fig. 12). The CRAs created team stories during the first semester to help them understand the meaning of climate change resilience and to help them connect with the issues their communities are facing. It was used as a final project grade. These stories were then shared in the second semester after-school program as part of the lessons given to after-school students in the same community as described in our methods section. Similar sentiments from CRAs indicate growth and confidence in communicating climate change, which helped prepare them for the third semester of community resilience outreach.

### 4.1.3 Objectives – Fall 2022 Climate READY After-school Program

During the fall semester (2022), CRAs learned to teach a four-lesson after-school program that brought students together to investigate the complex issue of climate resilience as a team (Table 2; Fig. 8(a-c). This provided a safe space for students to talk about the challenges they face in their own community. For example, the Lake Worth Beach team learned that one of their after-school students at Barton Elementary School experienced a heat stroke during school hours, earlier that school year. Although the after-school student was given immediate medical attention and he was ok and able to share his story, this was a traumatic event that affected the entire school population. Our CRAs used this personal story to discuss how climate change effects such as extreme heat can cause health problems like heat exhaustion or heat stroke and that the dangers are real for their community (The Fifth National Climate Assessment, 2023; Southeast Florida Regional Climate 2023 Compact, 2022; U.S. Global Change Research Program, 2016). When it was time to create a new climate resilient storybook, the Barton Elementary after-school students decided

to write about extreme heat, *Pablo Protects the Community from Extreme Heat* (see Appendix 2, Table 6), which addressed the program objective for fourth and fifth grade after-school students to "design and complete a storybook on community resilience to build an understanding of climate change and the impacts facing Southeastern Florida (Table 1). The personal experiences from the after-school students helped them consider solutions and with the help of the CRAs and the training they received through the Climate READY Program, after-school students chose to plant trees to create more shade, decreasing the heat island effect, and reducing risk of heat related illnesses. In their storyline, a gardener, a tree caretaker, and everyone from their community worked together to plant trees throughout the community. In the end, they celebrated with an ice cream party, and everyone was protected from the dangers of heat exhaustion and heat stroke. Even though the CRAs were the teachers in this setting, they were also learning from the elementary school students through this student community engagement activity, an important tool in training future leaders (Millican and Bourner, 2011).

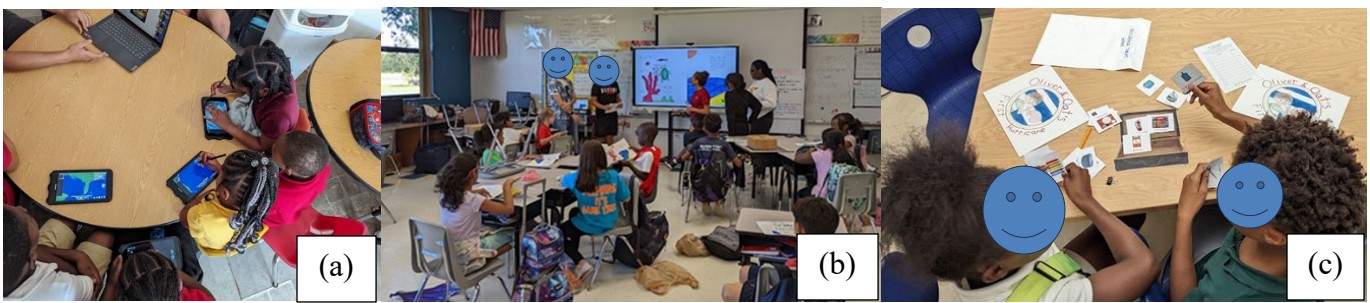

**Figure 8:** After-school students participating in the Climate READY Program. NOAA's Science on a Sphere and mobile application was used to help students understand climate change and how it relates to the effects seen in Florida (a). Climate READY Ambassadors were the teachers of 4 lessons including the creation of a storybook (b), and students were given a hurricane resilience game called "Get Storm Ready" to help them decide what might be the best items to pack in preparing for stronger storms (c).

Pre- and post- surveys were given to the after-school students before and after the four-lesson program (see Appendix 3, Tables 23-27). Sixty learners (N=60) completed the pre survey and 53 completed the post. The results indicate that significantly (p<0.05) more learners understood the science of the greenhouse effect, that humans are causing the current climate change we're witnessing, that extreme heat and rising seas are ways that climate change could affect South Florida, and they were able to identify ways to help reduce climate change (p<0.05) including walking or riding a bike in place of driving, unplugging TVs and computers when not in use, turning off lights when leaving a room, and restoring coastal habitats, the last being somewhat unexpected (see Appendix 2, Table 27). From classroom experience, many of our underserved youth in Palm Beach County have claimed to have never been to the ocean, let alone fully understand the dynamics of a coastline. It seems that their interactions with the CRAs helped them explore this ecosystem and they recognized that ecological restoration projects like planting native mangroves along our coastline would improve community resilience to sea level rise and other effects of climate change (Su et al. 2021). Our CRAs experienced this with the in-service project at the MANG nursery in the summer (Fig. 5) and they communicated their experience with the younger after-school students. These outcomes from both groups of students are seemingly very successful, where students demonstrated learning by doing and/or learning by storytelling (Maccanti et al., 2023; Lawrence and Paige, 2016).

### 4.1.4 Objectives – Spring 2023 Climate READY Community Outreach

Similar to other community-based approaches to address climate change (Clark et al., 2023; Semmens et al, 2023; McNamara and Buggy, 2017; Uitto and Shaw, 2006), the Climate READY Program reached out to the greater community during the spring semester. In the course *Community Resilience Outreach*, CRAs were given the task to generate a community resilience plan based

on what they learned throughout the program and to present their plan during a community event. While targeting the objectives to "identify and evaluate personal and community strengths and vulnerabilities in response to extreme weather events," "acknowledge that disproportionate distribution of vulnerabilities and diverse community values exist," and "design and implement community resilience-related service-learning projects based on local environmental challenges (Tabel 1), CRAs learned from local professionals throughout the program and developed community-specific resilience plans for each target community (Fig. 1). These plans were then shared with each community in at least two settings, one being a public presentation, and another could be a table event (see Appendix 2, Fig. 11). Once again, the CRAs were the teachers for community members, adults and families, addressing the objective "empower students to act as agents of change within the community by teaching community members about local climate impacts and resilience strategies for extreme weather events" (Table 1). An optional survey was conducted for those community members that attended a presentation session with our CRAs and we received 24 responses ranging from ages 10-15 to 51-60 (see Appendix 4, Table 31). Most were male (96%) and with varying levels of education (middle school to doctorate) (see Appendix 4, 33–34). Community members (N=24) rated the organization, quality, relevance, and usefulness of the presentation with high averages of 9/10 or above (see Appendix 4, Table 28). When asked "Did the interaction with the students change your thinking about community actions to address climate change? If so, how?" 23/24 of the responses were yes (see Appendix 4, Table 29). Several of the responses included "usually I don't think about things unless they affect me directly, so having people talk about it to us really opened my perspective," "I learned about legislation and landscaping," and "I think that having a younger person presenting gave us a better perspective."

The CRAs who participated in the table events were also well received, though a survey was not given. For example, our Lake Worth Beach team caught the attention of the current vice mayor, Christopher W. McVoy, a soil scientist and wetland expert, at an Earth Day Taco Fiesta event. Lake Worth Beach has not created an Office of Sustainability and/or Resiliency like the surrounding cities of Boca Raton, Boynton Beach, Delray Beach, and West Palm Beach, so the CRAs were captivated by his attention to climate change education, resilience, and their youth involvement in the community. Much of their discussion focused on the need for their community to have a sustainability office in Lake Worth Beach local government and what steps the youth could take to encourage Lake Worth Beach to work towards a more informed community. Dr. McVoy shared his contact information, and the CRAs were excited to continue the conversation. Part of the hope for this program was that local governments would give these students (CRAs) a platform to share their perspectives and research on environmental issues much like Boca Raton has done with a Youth Subcommittee (Youth subcommittee puts words into action in Boca Raton, 2019). The interaction with Dr. McVoy was an excellent first step towards involving youth in local government for Lake Worth Beach; therefore, meeting active local government officials proved to be an important aspect of the Climate READY Program. Another example where government officials interacted with our CRAs (outside of the course guest speaker encounters during class time) was through the Palm Beach County Office of Resilience (PBCOOR) and the urban sustainability directors network (USDN). The Director of PBCOOR, Megan Houston, and a Resilience and Sustainability Analyst, Natalie Frendberg, contacted our CRAs to participate in an USDN funded project, Strengthening Frontline Capacity for Climate Resilience Planning in Palm Beach County, FL (Final Report 2023). Ten of our CRAs from our Boca Raton, Lake Worth Beach, and Riviera Beach teams represented Palm Beach County Youth in presentations involving community members in Belle Glade, Pahokee, and West Palm Beach (Fig. 9A-C). These CRAs were able to play active roles in climate resilience planning for these communities, further addressing the program objective to "empower students to act as agents of change within the community by teaching community members about local climate impacts and resilience strategies for extreme weather events," and influencing their sense of community (Table 1; Fig. 9a-c).

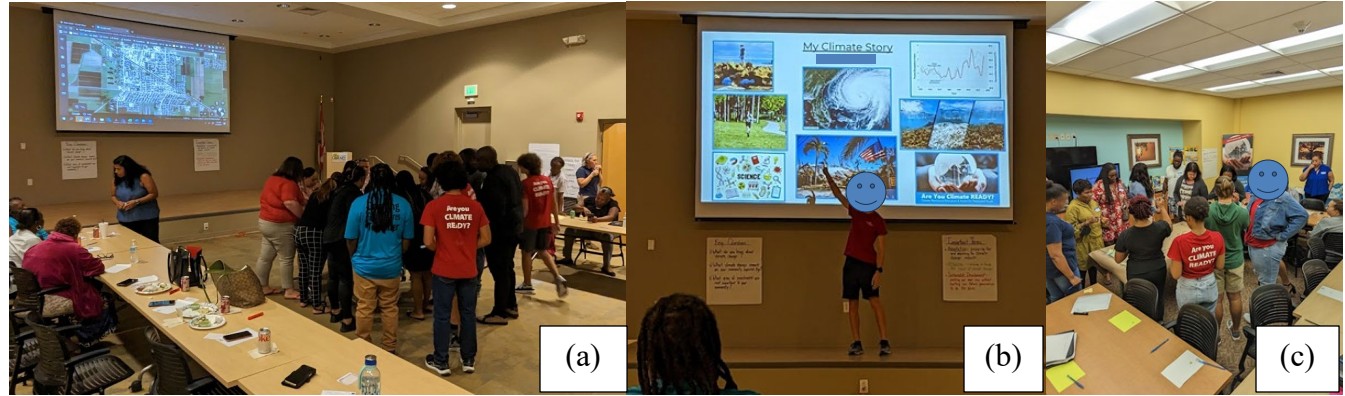

**Figure 9:** Climate READY Ambassadors (CRAs) with the Palm Beach County Office of Sustainability participating in the USDN funded project, Strengthening Frontline Capacity for Climate Resilience Planning in Palm Beach County, FL (Final Report 2023). Students travelled to Belle Glade and assisted in determining community assets (a) and shared their climate stories with community members (b). The CRAs also interacted with community members in West Palm Beach (c).

**4.2 Hypothesis Analysis**

Our hypothesis for the program was that participants in the Climate READY program will better understand their community strengths and vulnerabilities to a changing climate and that they will feel empowered to participate on both a personal and civic level to take action, minimize risks, adapt, and weigh the potential impacts of their decisions. We tested this hypothesis by designing and implementing a three-semester dual enrollment program and using pre- and post- questionnaires to compare means and identify significant differences ($p<0.05$). The mixed methods approach allowed us to focus on measuring changes to participant Behaviors, Attitudes, Skills, Interests, and Knowledge (BASIK), and proved to be an effective tool in telling our Climate READY story. Through evidence as discussed in this paper, high school students ages 15 to 17 that completed all 3 semesters and all pre- and post- questionnaires (N=23) were effectively taught climate change science and resilience (see Appendix 3, Tables 7-22), engaged in community settings to understand strengths and vulnerabilities (see Appendix Tables 7-9; Fig. 6, 8-9), and felt empowered to take action and be more involved in community decision making (Table 3; Fig. 9). In addition, the younger after-school students that took part in the four-lesson after-school part of the program and answered pre- (N=60) and post- (N=53) questionnaires demonstrated an increased understanding of climate change, community resilience, and how they could contribute to being an active community member in combatting the effects of climate change (see Appendix 3, Table23-27). Although our sample sizes are relatively low (below 100), we feel that our results of this study support our hypothesis. We plan to make efforts to continue testing these hypotheses by providing this three-semester model to more youth in the community if our budget allows. We have hopes to extend our dual enrollment (high school/university) opportunity to surrounding Broward, Miami-Dade, and Monroe Counties. Finding funding is always a challenge.

**4.3 Program Challenges and Recommendations**

The Climate READY program described in this paper encountered several challenges throughout the course of the grant funded period (October 2020 to September 2023). The largest challenge was to adapt our in-person model to a hybrid virtual one through the unexpected global pandemic (COVID-19) in 2020. The School District of Palm Beach County opened their schools for hybrid in-person and online learning throughout the 2020-2021 academic year where students and parents were given a choice (School District of Palm Beach County COVID-19 Guiding Document 2020 – 2021 School Year, 2020). All teachers and school staff were required to be onsite. Therefore, our Climate READY three-semester model also needed modifications as described in this paper to accommodate all eligible students, teachers, and after-school staff involved. Restrictions were lifted for the 2021 – 2022

academic year and all students were required to return to in-person schools, which proved to be another challenge for our community as everyone needed to learn new routines for a second time. We faced other challenges such as our struggle to recruit students in the more rural target communities of Pahokee and Belle Glade, and we underestimated the amount of staff time needed to perform all our planned tasks successfully. In the end, the support of our community from all levels (10–11-year-old after-school students to adult community members) gave us the strength and drive to complete our work.

Using our lessons learned from the Climate READY program, we recommend that future projects at FAU Pine Jog and institutions that plan to implement similar youth empowerment programs:

1. invest in quality staff and program coordination time, allowing for unexpected events such as global pandemics and natural hazards like hurricanes, earthquakes, flooding, etc.;

2. plan for multiple in-person recruitment meetings at several target schools, especially those in the most underrepresented and underserved areas; and

3. create and use an advisory council wisely to understand the real needs of the targeted community and stay in touch with them throughout the entire process with regular meetings. Our advisory council played a major role in our success.

## 5. Conclusions

The student Climate READY Ambassadors (CRAs) (N=23) in our three-semester long study have shown improvements in multiple areas including content knowledge of the causes, effects, and solutions of climate change, sense of place, and self-efficacy. They also built relationships with their communities. This program gave them a sense of empowerment to be agents of change to build community resilience to the effects of climate change. Our methods provided a safe space for members of the community to discuss important complex environmental issues. It also built skills for our youth to communicate to multiple stake holders such as government officials, resilience professionals, and the community at large. Students were inspired, motivated, and more likely to retain information while participating in this place-based climate resilience program. The three-semester experience at FAU Pine Jog gave students confidence to act and empowered them to be a part of their community decision making processes.

The fourth and fifth grade students (Pre N=60, Post N=53) in our study also showed significant improvements in their understanding of climate change and were able to identify ways to help reduce climate change. Several community members (N=24) took the time to answer questionnaires that indicated that the CR Ambassadors were effective in communicating community resilience to climate change. In addition, we estimate that over 700 community members were impacted during the three-semester study (Cohort 2), making a lasting impression among underserved regions of Palm Beach Couty, Florida (Fig. 2).

The Climate READY Program was a success and laid out a foundation for future climate resilience programming at FAU Pine Jog. It also provides an example of a youth empowerment program that could be shared and implemented with other colleges and universities. The data collected, experiences, and lessons learned from this study have given us the tools we need to move forward in our South Florida community to be more resilient to the challenges ahead and to be ready for the impacts of climate change.

Future research in the area of climate change education and youth empowerment could involve using this three-semester model or something similar at additional universities and institutions. Other geographical locations experience the effects of climate change differently than South Florida where concerns might be increased wildfires, shorter snow seasons, or increased drought. Curriculum could be tailored to each location given their specific concerns, strengths, and weaknesses, much like our Climate READY

Ambassadors discovered about their Palm Beach County communities. We need to be prepared and thus be more resilient to change. Our staff at FAU Pine Jog are always open to collaborations to community projects and would welcome ideas to improve our program as well share our resources with others. Climate change is a global issue and it will take a global village to be resilient.

**Appendix 1 – Glossary of terms**

**After-school students** – students ages nine to 11 enrolled in local community programs after their regular school day, often called aftercare programs, led by the Climate READY Ambassadors (ages 15-17) during the Fall Semester of the three semester Climate READY program. These students were also enrolled in local American elementary schools, grades four and five.

**Climate advocacy** – the act of advocating (publicly supporting) actions, policies, and various measures to address the effects of
climate change, improve public health, and to support a more sustainable society (Medoza-Vasconez et al., 2022).

**Climate READY** – Florida Atlantic University's Climate Resilience Education and Action for Dedicated Youth Program funded by the National Ocean and Atmosphere Administration Environmental Literacy Program (NOAA- NA20SEC0080016).

**Climate READY Ambassadors** – students ages 15 to 17 that were dual enrolled in the FAU Ping Jog Climate Resilience Education and Action for Dedicated Youth Program, and completed three semesters of training to be active communicators of
climate science and contributing community members towards environmental resilience and sustainability. These students were also enrolled in American secondary high schools, grades nine through twelve.

**Environmental education** – "a process that helps individuals, communities, and organizations learn more about the environment, develop skills to investigate their environment and to make intelligent, informed decisions about how they can help take care of it" (NAAEE, 2024).

**Environmental literacy** – part of a theoretical framework that focuses on the concern for the natural world, awareness of environmental problems, the ability to work towards solutions, and the knowledge and skills to prevent future environmental issues (Hollweg 2011; McBride et al., 2013).

**Ground zero** – a term often used by political and institutional leaders and the media to describe a place where changes are happening or where they begin to happen. In the context of climate change, this term is often used when describing the city of
Miami, Miami-Dade County, or the Miami Metropolitan Area that encompasses the southeastern region of Florida including Broward, Miami-Dade, and Palm Beach counties (Rifat and Liu, 2019); a highly populated area estimated at over six million people, a low lying metropolis vulnerable to sea level rise, intensifying storms, and extreme heat, and one of the lowest rated areas in disaster resilience in the United States (Rifat and Liu, 2020).

**LEED Certification** – Leadership in Energy and Environmental Design (LEED) is an United States green building rating system
ranging from "Certified" with 40-49 points to "Platinum" with +80 points. Point values are calculated based on how each project addresses building materials, carbon, energy, health and indoor quality, transportation, water, and waste (U.S. Green Building Council, Inc. - LEED rating system, 2024).

**Poverty line/level** – the United States federal government sets a federal poverty level that is an official measurement based on family size and income. Families that are at or below the distinction of poverty are able to receive government assistance programs
that help provide food, shelter, health care, and educational services (U.S. Department of Health and Human Services – Poverty Guildelines, 2024).

**School levels** – in the United States school system high school students in ninth through twelfth grades, considered as freshman, sophomores, juniors, and seniors, range in age of fourteen to eighteen. Students in elementary school are in kindergarten to fifth grades, range in ages of five to eleven.

**Title 1 Schools –** schools that receive financial assistance through a United States Federal grant program established by The Elementary and Secondary Education Act and The Every Student Succeeds Act. These schools have a high percentage of children at or below poverty level and qualify to receive supplemental funding to assist children in meeting "challenging state academic standards" (National Center for Education Statistics, 2019).

**Appendix 2 – Figures and tables of curriculum used**

FAU Pine Jog selected a sequence of 10 NOAA SOS datasets to create a playlist for the students. This playlist was an important part of teaching climate change science as it provided visual representations of global carbon dioxide concentrations, global temperature, global hurricane pathways, and changes in arctic sea ice coverage over time. Video shorts created by NOAA partners were also used.


**Table 5:** Playlist of NOAA SOS datasets used in order of presentation. Source Data: https://sos.noaa.gov/Datasets/

| | Name | Data Type | Duration | Notable Features |
|---|---|---|---|---|
| 1. | Blue Marble | Satellite Earth | NA | <ul><li>Vastness of the Sahara Desert</li><li>Shading done in true color: gives Earth's appearance from space</li></ul> |
| 2. | Changing Climate, Changing Ocean | Movie | 6:30 | Focuses on impacts of CO2 on climate and ocean health and need for action. |
| 3. | Carbon Dioxide Concentration: GEOS-5 Model | CO2 Model | NA | <ul><li>In North America, notice how weather patterns affect carbon dioxide distribution in the atmosphere. Emissions in the U.S. Midwest and East Coast are carried east by the westerly winds to the Atlantic Ocean.</li><li>In Asia, major emissions in industrialized Asian countries are apparent and move eastward.</li><li>In Africa, plumes of white, carbon monoxide emissions, are seen from fires.</li></ul> |
| 4. | Climate Model: Temperature Change CCSM b1 1870-2100 | Temp Model | NA | The temperatures displayed in the datasets are all a comparison to temperatures in 2000. Blue tones on the visualization represent temperatures cooler than those in 2000, while red tones represent temperatures warmer than those in 2000. |
| 5. | Sea Ice Extent-1978 - Present | Satellite Sea Ice Data | NA | <ul><li>Seasonal change of sea ice</li><li>Shrinking of Arctic sea ice concentration, especially in summers</li><li>The disappearance of the Odden, a thumb-shaped sea ice feature east of Greenland, which often is visible prior to the late 1990's</li><li>The minimum sea ice concentration in 2007 shattered the previous minimum sea ice record set in 2005 by 23% and contained 39% less ice than the 1979 to 2000 average.</li><li>The minimum sea ice extent record was broken again in September 2012</li></ul> |

| | | | | |
|---|---|---|---|---|
| **6.** | Rising Sea | Movie | 6:23 | Scientists estimate sea levels could rise more than three feet by the end of this century. That would mean the flooding of commercial and residential property along the coast. |
| **7.** | Hurricane Tracks: Cumulative 1950-2020 | Satellite Hurricane Data | NA | • All recorded hurricanes worldwide from 1950 - 2020 are included<br>• No hurricanes cross the equator<br>• Very few hurricanes make it to South America because of wind shear patterns. |
| **8.** | Hurricane Season 2020 | Satellite Hurricane Data | | • 30 named storms, of which 13 became hurricanes and 6 of those became major hurricanes<br>• Fifth consecutive above-normal year for hurricane activity<br>• 10 storms formed in September alone<br>• 12 storms hit the US coastline |
| **9.** | Resilient Cities: Key to Thriving on a Changing Planet | Movie | 4:17 | • Cities have long been crucibles of creativity, innovation and wealth generating engines. Ex. Athens, Rome, Delhi and Peking<br>• More than 50% of the global population live in cities, will reach 70% by 2050 marking the largest migration of humans in history.<br>• Cities provide rich and unusual opportunities to reduce the human ecological footprint on Earth and to leave more open space for nature… but seizing these opportunities requires that our cities become denser, greener, and smarter. |
| **10.** | Holocene/Little Boxes | Movie | 5:48 | • A scientifically responsible, arts-expressed film with visceral imagery, a succinct narrative, and global data that invites us to reflect on our relationship to our changing climate and rising levels of CO2.<br>• Everyday behaviors are translated into a visual metaphor about the complex and connected relationship between our actions and ways of being in the Anthropocene and our ever-changing climate. |

Students were asked to write a "Where I'm From" poem to evaluate self-awareness and sense of place early in the summer 2022 CR Ambassador Institute (Semester 1). They were then asked to illustrate their poems on a paper puzzle piece that was later placed 865 together in a community puzzle with their fellow CR Ambassadors at the end of the course (Fig. 10).

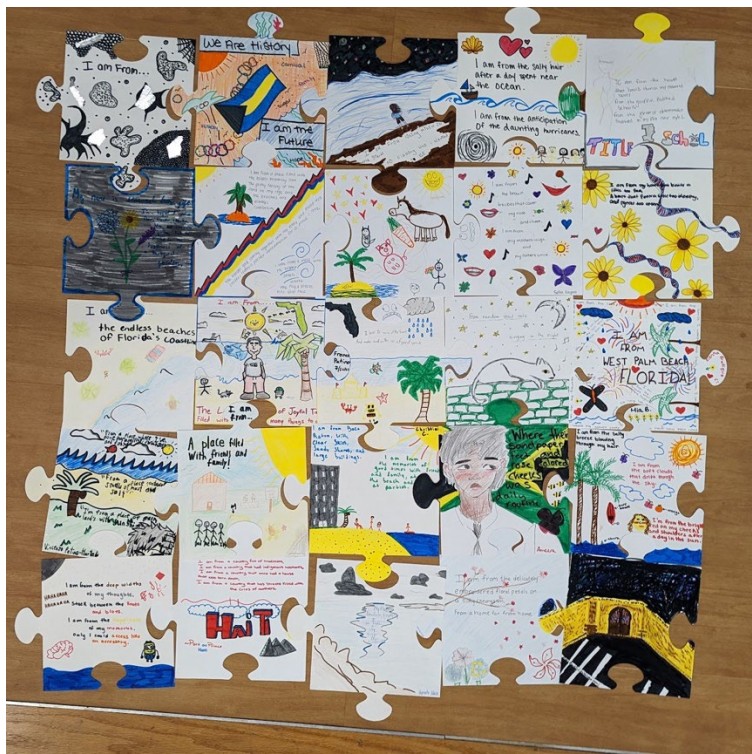

**Figure 10:** Community puzzle created by Climate READY Ambassadors during the summer 2022 CR Ambassador Institute (Semester 1).

Scheduling and time management assignments were very important to the success of the Climate READY program. Shared calendars were used in the fall and spring semesters to help students and instructors manage their time efficiently (Fig.11). Students were given the opportunity to choose what was best for them, as long as they communicated their needs in a timely matter. This was important because all CR Ambassadors had a full-time high school schedule in addition to the FAU dual enrolment course. Many of them were also working part-time jobs and/or involved in extracurricular clubs and sports.


**Master Schedule of Graded Outreach Events Spring 2023**

| Sunday | Monday | Tuesday | Wednesday | Thursday | Friday | Saturday |
|---|---|---|---|---|---|---|
| | | | | | | Feb 25 (Class #4) |
| Feb 26 | Feb 27 **Symphonia** 5pm - 8pm | Feb 28 | March 1 | March 2 | March 3 | March 4 |
| March 5 | March 6 | March 7 | March 8 | March 9 | March 10 | March 11 |
| March 12 | March 13 | March 14 | March 15 | March 16 | March 17 | March 18 |
| March 19 | March 20 | March 21 | March 22 | March 23 | March 24 | March 25 (Class #5) |
| March 26 | March 27 | March 28 | March 29 | March 30 | March 31 | April 1 **WPB e4 Life @Cox** |

| | | | | | | |
|---|---|---|---|---|---|---|
| | | | | | | **Science** <br> 10am - 4pm |
| April 2 | April 3 | April 4 | April 5 <br> **Summit Library** <br> 6 - 7:30p | April 6 <br> **Lantana Rd. Library** <br> 6-7:30pm | April 7 | April 8 |
| April 9 | April 10 | April 11 | April 12 | April 13 | April 14 | April 15 |
| April 16 | April 17 | April 18 <br> **Boynton Beach Library** <br> 5:30-7pm | April 19 | April 20 <br> **Max M. Fisher Boys & Girls Club** <br> 4:30-6pm | April 21 | April 22 <br> **WPB in Okeeheelee** <br> 10am - 3pm <br> **Taco Fiesta in Lake Worth** <br> 3-7pm <br> **Boynton Earth Day** <br> 10am-1pm |
| April 23 | April 24 | April 25 | April 26 | April 26 | April 28 | April 29 <br> (Class #6) <br> In person <br> @ Pine Jog <br> 8:30 to 5pm |

Key

| | |
|---|---|
| Anne <br> Rachel <br> Graduate Student | Adult Supervision Schedule |
| | Community Outreach Events |
| | Regular Class Time |


**Figure 11:** Schedule of outreach events during semester 3 of the Climate READY Program (Spring 2023), Community Outreach. Events began February 27 and ended on April 22 with the last class meeting on April 29. Note: Four of the 13 graded outreach events were not on this student schedule as they were planned and implemented prior to this Time Management assignment. Those events included 3 Urban Sustainability Directors Network (USDN) grant collaborations with the Palm Beach County Office of
Resilience (Amplifying Impact Community Partners of South Florida, 2023) and a Climate and Art event coordinated by the Delray Beach Office of Sustainability and Resilience (Climate and Art Delray Beach 2022).

Ten original stories were created by student authors during cohort 2 of Climate READY; 5 were created by the Teen Ambassadors and 5 were created though the collaboration with the CR Ambassadors and the fourth and fifth grade students at 5 different Title 1 after-school programs (Table 9). Each original storybook was printed using the self-publishing and book printing company,
DiggyPOD (2004). Copies of each storybook were distributed to student authors and any remaining copies were given to the school's administrators to distribute to their main library or individual classroom libraries. Access to these storybooks (pdf) can be made available upon request from the authors of this paper.


**Table 6:** Original community climate resilience stories created by Teen Ambassadors and fourth and fifth grade after-school students in Title 1 schools during cohort 2.

| Palm Beach County Community | Storybook Title | Student front cover art | Student Authors |
|---|---|---|---|
| Boca Raton | Sandy Saves the Sea Shore |  | Boca Raton CR Ambassadors |
| Boynton Beach /Delray Beach | Shelly's Surprisin Shore Story |  | Boynton Beach /Delray Beach CR Ambassadors |
| Lake Worth Beach | Oliver & Oat's First Hurricane |  | Lake Worth Beach CR Ambassadors |
| Riviera Beach | Sally's Sea Adventure |  | Riviera Beach CR Ambassadors |
| West Palm Beach | Millie Through Metamorphosis: The Changing World |  | West Palm Beach CR Ambassadors |
| Boca Raton | Miami Manages Mangroves |  | Boca Raton CR Ambassadors and the fourth and fifth grade after-school students at Coral Sunset Elementary School |

| | | | |
|---|---|---|---|
| Boynton Beach /Delray Beach | Star's Hurricane Plan |  | Boynton Beach /Delray Beach CR Ambassadors and the fourth and fifth grade after-school students at Galaxy E3 Elementary School |
| Lake Worth Beach | Pablo Protects the Community from Extreme Heat |  | Lake Worth Beach CR Ambassadors and the fourth and fifth grade after-school students at Barton Elementary School |
| Riviera Beach | Mark's Climate Adventure |  | Riviera Beach CR Ambassadors and the fourth and fifth grade after-school students at University Learning Academy |
| West Palm Beach | Joquin and Peanut B. Save the Coast |  | West Palm Beach CR Ambassadors and the fourth and fifth grade after-school students at Pine Jog Elementary School |


Climate READY Ambassador students answered free responses questions with more detail in the post 1 questionnaire and provided shorter more concise answers in the post 2 questionnaire (Table 5.12 and Table 8). The top three responses were (1) seal level rising, (2) extreme heat, and (3) stronger storms or hurricanes.

**Community Resilience Storybook Outline**

Create the storyline for the storybook about community resilience. This assignment will help you prepare your storyline for your community group where you will have chosen a **main character**, a **climate impact** (heat, sea-level rise, or hurricanes/storms), an **action or solution**, and **three helpers** from your community. You will now use those ingredients to write the narrative for your story, page by page, on this template.

**Which story components did your students select?**

1.  **Main Character:**

2.  **Climate Impact:**

3.  **Action/Solution:**

4.  **Three Helpers (People in the community with knowledge and skill to help):**

    a. ______________________________________________

    b. ______________________________________________

    c. ______________________________________________

ASSIGNMENT: Using the story ingredients above, follow the page-by-page prompts (below) to write 1-2 sentences for each page of the story. (These are the lines that will be printed on the pages of the class book.) When you are finished, it should sound like a story! Read through each "page" you have written to make sure it sounds smooth and complete.

**Figure 12:** FAU Pine Jog Original Community Resilience Storybook Outline assignment for dual enrolled CR Ambassadors and fourth and fifth grade students in the Climate READY Program.

**Appendix 3 – Survey responses from Climate READY Ambassadors (CRAs) and after-school students**

**Tables 7–9**: Climate READY Ambassador student perception of community vulnerability, why people question climate change, and the science of climate change.

**Table 7**: Do you think your community is already being affected by climate change?

|                    | Pre  | Post 1 | Post 2   |
|--------------------|------|--------|----------|
| Not at all         | 0    | 0      | 0        |
| Only a little      | 5%   | 9%     | 5%       |
| A moderate amount  | 41%  | 64%    | 30%      |
| A great deal       | 50%  | 27%    | **65%**** |
| Don't know         | 5%   | 0      | 0        |

**Indicates a significant difference between Post 1 and Post 2 ($p < 0.05$)

**Table 8**: What do you think is the main reason that people question the science of climate change?

|                                          | Pre  | Post 1 | Post 2   |
|------------------------------------------|------|--------|----------|
| A lack of science knowledge on the topic | 36%  | 32%    | **55%**** |
| A lack of trust of scientists            | 5%   | 18%    | 15%      |

| | | | |
|---|---|---|---|
| The media presenting "both sides" of the issue | 14% | 5% | 10% |
| Political or religious background | 36% | 41% | **20%**\*\* |
| Other | 9% | 5% | 0 |

**\*\*Indicates a significant difference between Post 1 and Post 2 (p<0.05)**

**Table 9**: What causes YOU to question the science of climate change?

| | Pre | Post 1 | Post 2 |
|---|---|---|---|
| A lack of science knowledge on the topic | 23% | 9% | 10% |
| A lack of trust of scientists | 0 | 5% | 0 |
| The media presenting "both sides" of the issue | 9% | 0 | 5% |
| Political or religious background | 5% | 5% | 0 |
| I do NOT question the science of climate change | 64% | 82% | **85%**\* |

**\*Indicates a significant difference pre to post (p<0.05)**

**Tables 10–22:** Climate READY Ambassador student responses to climate change content knowledge questions. Significant differences are identified between pre- and post- questionnaires (\*p<0.05) and between Post 1 and Post 2 questionnaires (\*\*p<0.05). Answers graded as correct have a check mark to the left of the statement.


**Table 10:** If the greenhouse effect is natural, then why is today's climate change a bad thing? (Select all that apply)

| | % Response | | |
|---|---|---|---|
| | Pre | Post 1 | Post 2 |
| ✓ A small increase in greenhouse gas concentration can have a large effect of increased warming. | 59% | **91%**\* | 100% |
| ✓ Humans have altered a natural process and exaggerated changes that might normally occur over millions of years. | 68% | **86%**\* | 100% |
| ✓ Once released into the atmosphere, greenhouse gases remain potent for many years, making it difficult to reverse the process. | 50% | **91%**\* | 100% |
| The use of aerosols and other pollutants from human activities has created a hole in the ozone layer, allowing more heat to enter the earth's atmosphere and amplifying the greenhouse effect | 95% | **73%**\* | 85% |
| ✓ Abrupt changes to the climate system may have unintended outcomes that may pose challenges for societies, like more extreme weather, spread of diseases, a decline in marine life, or an alteration of ocean circulation patterns. | 68% | **86%**\* | 100% |

**\*Indicates a significant difference Pre to Post (p<0.05)**

**Table 11:** Which of the following is the major concern about climate change if CO2 levels in the atmosphere exceed 450 ppm?

| | % Response | | |
|---|---|---|---|
| | Pre | Post 1 | Post 2 |
| The earth would erupt into a huge fireball | 5% | 0 | 0 |
| ✓ The effects would be irreversible | 91% | 100% | 100% |
| All life on the planet would die | 5% | 0 | 0 |

| Some oil companies go out of business | 0 | 0 | 0 |

 *Indicates a significant difference Pre to Post ($p<0.05$)

**Table 12:** What are the 2 possible approaches to responding to a changing climate according to NASA and other scientists?

| | % Response | | |
|---|---|---|---|
| | Pre | Post 1 | Post 2 |
| Watching and waiting | 0 | 0 | 0 |
| Watching and mitigating | 9% | 0 | 0 |
| Waiting and adapting | 9% | 5% | 0 |
| ✓ Adapting and mitigating | 82% | 95% | **100%*** |

*Indicates a significant difference Pre to Post ($p<0.05$)

**Table 13:** Which carbon sequestration strategy has the highest probability of success?

| | % Response | | |
|---|---|---|---|
| | Pre | Post 1 | Post 2 |
| ✓ Restoring natural forests | 73% | **100%*** | 100% |
| Converting $CO_2$ to $MgCO_3$ | 23% | 0 | 0 |
| Injecting $CO_2$ into beneath the ocean floor | 5% | 0 | 0 |
| Mirrors in orbit | 0 | 0 | 0 |

*Indicates a significant difference Pre to Post ($p<0.05$)

**Table 14:** How fast do we need to stop burning fossil fuels to limit global temperature rise to 2 degrees C? (3.6 degrees F)

| | % Response | | |
|---|---|---|---|
| | Pre | Post 1 | Post 2 |
| We need to stop burning fossil fuels by 2100 | 23% | 5% | 0 |
| ✓ We need to stop burning fossil fuels by 2040 | 64% | **95%*** | 100% |
| Fossil fuels don't matter, the Sun will cool and so will the Earth | 0 | 0 | 0 |
| It's already too late to stay below the 2-degree threshold. We should have stopped burning fossil fuels in the early 2000s | 14% | 0 | 0 |

*Indicates a significant difference Pre to Post ($p<0.05$)


**Table 15:** Which two human activities are largely responsible for the observed atmospheric warming?

| | % Response | | |
|---|---|---|---|
| | Pre | Post 1 | Post 2 |
| Burning of fossil fuels and growth of urban areas | 22% | 9% | **50%**** |
| ✓ Burning of fossil fuels and clearing of forest land | 73% | **91%*** | **50%**** |
| Cigarette smoking and the explosion of airline travel | 0 | 0 | 0 |
| Burning of rainforests and loss of ice in the Arctic | 5% | 0 | 0 |

*Indicates a significant difference Pre to Post ($p<0.05$)

**Indicates a significant difference between Post 1 and Post 2 ($p<0.05$)

**Table 16:** How much has CO2 in the atmosphere increased since the Industrial Revolution? In the 10,000 years before the Industrial Revolution in 1751, carbon dioxide levels rose less than 1 percent. Since then, they've risen by:

| | % Response | | |
|---|---|---|---|
| | Pre | Post 1 | Post 2 |
| 11% | 9% | 0 | 5% |
| ✓ 43% | 50% | **95%*** | **45%**** |
| 62% | 36% | 0 | **30%**** |
| 75% | 5% | 5% | 20% |

*Indicates a significant difference Pre to Post (p<0.05)

**Indicates a significant difference between Post 1 and Post 2 (p<0.5)


**Table 17**: When was the last time in Earth's history that CO2 was as high as it is now?

| | % Response | | |
|---|---|---|---|
| | Pre | Post 1 | Post 2 |
| This is the highest it's ever been | 45% | 0 | **30%**** |
| CO2 was at least this high during the warm periods between the ice ages | 18% | 5% | 10% |
| CO2 has not been this high for almost one million years. | 14% | 5% | 0 |
| ✓ The last time CO2 was this high was 3 million years ago. | 23% | 91%* | **60%**** |

*Indicates a significant difference Pre to Post (p<0.05)

**Indicates a significant difference between Post 1 and Post 2 (p<0.05)


**Table 18:** Modern instruments have only been around for a little over 100 years. So how do we know what greenhouse gas concentrations (and temperature) were in Earth's past? (select all that apply) Answers graded as correct have a check mark to the left of the statement.

| | % Response | | |
|---|---|---|---|
| | Pre | Post 1 | Post 2 |
| ✓ Air bubbles trapped in ice cores provide detailed records of what the atmosphere was like in the past. | 50% | **95%*** | 95% |
| ✓ Examining organisms in marine sediments can tell us what the temperature was like in the past. | 59% | **73%*** | **90%**** |
| ✓ Pollen in lake beds shows what plant species have lived there during different times. Different plant populations are associated with different types of climates. | 45% | **68%*** | 85% |
| ✓ Glacial moraines show when and where previous episodes of glaciation occurred. | 86% | 86% | 90% |
| ✓ Tree rings show the history of drought, fire, and other environmental variations. | 59% | **86%*** | 95% |

*Indicates a significant difference Pre to Post (p<0.05)

**Indicates a significant difference between Post 1 and Post 2 (p<0.05)

**Table 19**: How long does CO2 remain in the atmosphere?

| | % Response | | |
| --- | --- | --- | --- |
| | Pre | Post 1 | Post 2 |
| CO2 washes out of the atmosphere seasonally. | 0 | 0 | 0 |
| CO2 remains in the atmosphere for 5-10 years. | 18% | 0 | 20% |
| ✓ CO2 remains in the atmosphere for up to 200 years, or more. | 41% | **100%*** | **60%**** |
| No single lifetime can be defined for CO2 because of the different rates of uptake by different removal processes. | 32% | 0 | 20% |

*Indicates a significant difference Pre to Post ($p < 0.05$)

**Indicates a significant difference between Post 1 and Post 2 ($p < 0.05$)


**Table 20**: What are the major causes of sea level rise? (There may be more than one correct answer)

| | % Response | | |
| --- | --- | --- | --- |
| | Pre | Post 1 | Post 2 |
| Melting sea ice | 73% | 86% | 95% |
| ✓ Melting glaciers and ice sheets | 100% | 100% | 100% |
| Rivers accelerating | 9% | 5% | 5% |
| ✓ Seawater expanding as it gets warmer | 18% | **59%*** | **100%**** |

*Indicates a significant difference Pre to Post ($p < 0.05$)

**Indicates a significant difference between Post 1 and Post 2 ($p < 0.05$)

**Table 21**: Ranked tallies of repeated answers to the Climate READY Ambassador student free response question to "Can you identify some of the impacts of climate change that South Florida is likely to experience within the next 30 years? Please list them here." Full responses are found in Table 27 under Appendix 3.

| Rank | Tally | Post 1 Response |
| --- | --- | --- |
| 1 | 20 | rising sea levels |
| 2 | 13 | increasing to extreme temperature |
| 3 | 11 | hurricanes |
| | 8 | flooding |
| | 6 | beach erosion |
| | 5 | decline of native organisms |
| | 3 | loss of habitats and ecosystems |
| | 3 | coral bleaching |
| | 3 | more extreme natural disasters |
| | 2 | more urban heat islands |
| | 2 | the cost of living can increase |
| | 2 | ocean acidification |
| | 2 | agriculture can be damaged |
| | 2 | salt-water intrusion |

| Tally | Post 2 Response |
| --- | --- |
| 19 | Sea level rising |
| 15 | extreme heat |
| 13 | stronger storms |
| 6 | increased urban island effects |
| 6 | Loss of habitat |
| 4 | coastal erosion |
| 3 | flooding |
| 3 | extinction of native species |
| 2 | greenhouse effect |
| 1 | increased insurance prices |
| 1 | hotter summers and cooler winters |
| 1 | ocean acidification |
| 1 | saltwater intrusion |

| | 2 | overpopulation |
|---|---|---|
| | 1 | toxic algae blooms |
| | 1 | population decrease |
| | 1 | melting glaciers |
| | 1 | I don't know |
| | 1 | human migration |
| | 1 | extinction of certain species |
| | 1 | heat-related illnesses |

**Table 22:** Detailed Climate READY Ambassador student free response answers to "Can you identify some of the impacts of
climate change that South Florida is likely to experience within the next 30 years? Please list them here." Student responses were
more detailed and factual during Post 1 and more concise in Post 2 (see Table 19 for ranked tallies of student responses).

**Post 1 Student Responses**

- Some impacts of climate change that will take place in South Florida include rising sea levels, increasing to extreme temperature, loss of habitats and ecosystems, increased opportunity for flooding, hurricanes and storms with increased strength, more urban heat islands, and decline of native organisms.
- Rising sea level can lead to flooding and damage, the intensifying heat can make living conditions dangerous especially for elderly and low-income communities, the cost of rent can increase as more people move inland, our oceans can face coral bleaching and acidification, our crops and agriculture can also become damaged.
- Rising sea level, coral reef bleaching, more intense hurricanes
- Some impacts of climate change that South Florida is likely to experience is high sea level rise, beach erosion and flooding and also there is a likely chance that South Florida could be going through extreme storms, hurricanes within the next 30 years.
- Sea level rise, extreme heat, more extreme natural disasters, toxic algae blooms, population decrease, increase in prices for resources, loss of coral reefs, coastline erosion.
- Within the next 30 years Florida will see sea levels-rising due to climate change and the melting glaciers and we will see temperate rising, and more heat waves; which can cause more horrific hurricanes and other natural disasters.
- -rising sea levels
  -increasing temperatures
  -salt-water intrusion
  -more impactful natural disasters
- idk
- Concerning Florida, within the next 30 years sea level rise will raise a multitude of impacts on South Florida. The rising of our oceans due to global warming can cause shoreline erosion, habitat destruction of animals and humans, as well as plant life across Florida's shores.
- flooding due to rising sea levels, extreme heat, and extreme hurricanes
- Florida is likely to experience extreme heat, sea-level rise, land going underwater, and more.
- Sea level rise, worsening hurricanes, ocean acidification, flooding, mangroves dying, etc.
- South Florida will experience sea level rising, slat water intrusion, more severe storms, and heat index rising. All of these issues will harm the thousands of people who live on the coast as well as the agriculture industry here. Honestly, they will affect everyone (not equally) in some shape of form. Our coastlines and natural habitats are already being eroded as floods increase.
- Sea levels rising into our neighborhoods, loss of wildlife including mangroves, human migration
- South Florida is likely to experience flooding, rising sea levels, and more severe storms (hurricanes).

- Rising sea levels, Higher coast erosion, Stronger and more common hurricanes, Urban Heat Islands, Increasing overall temp
- Increased temperatures
  - Rising sea levels
  - Overpopulation
  -Extinction of certain species
- Some of the major impacts that South Florida is likely to experience within the next 30 years due to Climate Change include increases in overall temperature and heat, more intense weather and storms, and rising sea levels. An increase in temperature can result in hospitalities due to heat-related illnesses, while more intense weather and storms can lead to home destruction and the harming of local ecosystems. Finally, rising sea levels can cause the submersion of residences near the coast of Florida and can result in increasing soil erosion.
- Some of the impacts that South Florida is likely to experience in the next 30 years is extreme heat, rising sea-levels, and the exponential growth in world population. Sea levels may rise to about 3 feet, the average temperature will rise up about 1 degree Celsius, and the global population would be too overwhelmed from an incoming carrying capacity.
- Rising sea levels,
  Temperature increase,
  eroding beaches,
  increased intensity in storms
- sea level will rise harming the community and wildlife.
- Seal Level Rise, Flooding, Severe Hurricanes

**Post 2 Student Responses**

- Sea level rising, increased insurance prices, stronger storms, hotter summers and cooler winters, and increased precipitation.
- Sea level rise, worsening storms, ocean acidification, coastal erosion, extreme heat, etc
- rising sea levels increasing temperatures flooding strengthened natural disaster (hurricanes)
- Rising seas level, increased urban island effects, extinction of native species, sealine corrosion, stronger storms
- Rising sea levels, stronger storms/ hurricanes.
- Sea level rise, stronger hurricanes from extreme heat and the greenhouse effect(greenhouse gases and more evaporation leading to stronger storms), permafrost melting and causing pathogenic diseases to come out from years ago, biodiversity decreasing including us
- Sea level rise, extreme heat, loss of habitat, stronger storms
- Stronger Storms, warmer temperatures, sea level rise, and the urban heat island effect
- Sea level rise Stronger storms Increasing temperatures Shoreline erosion Urban island heat Loss of habitat
- Sea level rise Urban heat islands Global warming
- Increase temperature Sea Level Rise Amplified storms
- Increased temperatures -Sea level rise -Increased heat waves -Increased storm strength, frequency, and duration
- extreme heat - rising sea levels - strengthening storms and hurricanes - loss of ecosystems - coastal erosion
- Rising sea levels, urban heat islands, general higher heats, habitat loss
- Sea level rise and high temperatures.
- Rising sea levels, increasing temps, loss of habitat
- Rising sea levels and increasing heat.
- Extreme heat, salt water intrusion, hurricane, floods, etc
- Rising sea levels, worsening storms, increasing temperature, harming of both terrestrial and marine ecosystems, etc.
- sea level rise urban heat island effect loss of habitat flooding

**Tables 23–27:** Pre- (N=60) and post- (N=53) questionnaire results from fourth and fifth grade after-school students, ages nine to 11. Significant differences were found (*p<0.05) using a Student's T-test with normal distribution. Answers graded as correct are highlighted in yellow.

**Table 23:** What do you think is causing our current climate change?

|  | % Chosen | |
| --- | --- | --- |
|  | Pre | Post |
| I think it is part of a natural cycle | 45% | **28%*** |
| I think people are causing it | 37% | **58%*** |
| I do not think the climate is changing | 8% | 2% |
| Don't know | 10% | 11% |

*Indicates a significant difference pre to post (p<0.05)

**Table 24:** True or False

|  | % Correct | |
| --- | --- | --- |
|  | Pre | Post |
| ✓ Climate means average weather conditions in a region over time. | 77% | 79% |
| Climate and weather are pretty much the same thing. | 75% | 81% |
| Weather is usually expressed in terms of temperature, precipitation, and wind. | 91% | 83% |

**Table 25:** The greenhouse effect refers to:

|  | % Chosen | |
| --- | --- | --- |
|  | Pre | Post |
| The Earth's protective ozone layer | 22% | 19% |
| Pollution that causes acid rain | 17% | 8% |
| ✓ Gasses in the atmosphere that trap heat | 24% | **60%*** |
| How plants grow | 17% | 10% |
| Don't know | 20% | **4%*** |

*Indicates a significant difference pre to post (p<0.05)

**Table 26:** What ways could climate change affect south Florida?

|  | % Chosen | |
| --- | --- | --- |
|  | Pre | Post |
| ✓ Extreme heat | 55% | **74%*** |
| ✓ Rising seas | 57% | **72%*** |
| Shorter days | 13% | 8% |
| ✓ Stronger storms | 55% | 66% |
| Fewer hurricanes | 25% | 23% |

*Indicates a significant difference pre to post (p<0.05)

**Table 27:** Which of the following actions can people take to help reduce the impacts of climate change?

| | | % Chosen | |
|---|---|---|---|
| | | Pre | Post |
| ✓ | Walk or bicycle instead of drive | 65% | **91%*** |
| ✓ | Unplug TVs and computers when not in use | 60% | **77%*** |
| ✓ | Turn off the lights when leaving the room | 65% | **87%*** |
| ✓ | Don't waste food | 47% | 43% |
| ✓ | Put houses on stilts | 10% | 23% |
| ✓ | Restore coastal habitats | 38% | **55%*** |
| ✓ | Make a family emergency plan | 25% | 28% |
| ✓ | Eat less meat | 13% | 9% |

*Indicates a significant difference pre to post ($p < 0.05$)


**Appendix 4 – Survey responses of community members after learning from Climate READY Ambassadors**

Pre- and post- surveys were conducted before and after the after-school 4-lesson program to fourth and fifth grade students during the fall semester. These were not matched pairs where 60 learners completed the pre survey and 53 learners completed the post; therefore, percent were used for discussion (Table 9). An optional single post survey was conducted after the community outreach presentations in the spring semester. Twenty-four community members completed the survey across all presentations and mean responses were used for discussion (Tables 10.1 – 10.7).

**Tables 28–34:** Post presentation evaluation from community members in Palm Beach County (N= 24).

**Table 28**: Please rate the different aspects of today's presentation using a scale from 1-10 where 10=highest

| | Mean Response (N=24) |
|---|---|
| Organization and clarity of the presentation | 9.3/10 |
| Quality of information shared | 9.6 |
| Relevance to my community | 9.8 |
| Provided community actions for extreme weather and other environmental hazards | 9.0 |
| Usefulness of resources | 9.3 |

**Table 29**: Free response question on community perception of actions to address climate change.

| **Did the interaction with the students change your thinking about community actions to address climate change? If so, how? Yes = 23/24 (96%)** |
|---|
| • YES because usually I don't think about things unless they affect me directly, so having people talk about it to us really opened my perspective. <br> • Yes, it opened my perspective to our local vulnerabilities. <br> • Yes, it made me more aware of the environment that I live in. <br> • Yes, I learned a lot about my area's relationship to the climate. <br> • Yes by making me realize I can be doing more to prevent pollution. <br> • Yes, it gave me more clarity on what my community is doing to help climate change and what they should do conversely. <br> • Yes, I learned more about what we can do |

- I like how they presented ways to live sustainable such as through transportation and using energy efficiently
- I do, I had a more in-depth understanding
- No, because I already knew all of the information.
- Yes, I think that having a younger person presenting gave us a better perspective
- Yes, more ideas about how to be environmentally conscious
- Yes, I didn't realize the aspects our community had on the overall climate and the way all the parts fit together are really interesting
- Though the PowerPoint was very thorough, I already have these things on my mind however, the one fact about the potential loss of jet streams was extremely insightful and worrying
- Yes, I became more aware of how my community is helping climate change, inspiring me to get involved.
- They were fantastic!
- Yessir did it inspired me to talk and find more information on the subject
- Yes, I learned about the importance of planting trees
- Glad to see youth so engaged.
- Yes there is more local initiatives than I thought
- Yes, I learned about legislation and landscaping.
- Yes more urgent

**Table 30**: Please state your level of agreement with the following statements.

| | Mean Response | % Agree and Strongly Agree |
|---|---|---|
| I have a stronger understanding of the causes and impacts of climate change. | 4.4/5 | 92% |
| I have a better understanding of our community's vulnerabilities to extreme weather events and other environmental hazards. | 4.4 | 96% |
| I am more aware of our community's strengths and potential solutions in response to extreme weather events and other local environmental hazards | 4.5 | 100% |
| I am more aware of local and regional efforts to assess our community vulnerabilities and make our community more climate resilient | 4.5 | 96% |
| I feel I can contribute to helping our community be more resilient | 4.5 | 100% |


**Demographics of community members**

**Table 31**: Respondent age

| | N | % |
|---|---|---|
| Under 10 | 0 | 0 |
| 10-15 | 6 | 25% |
| 16-20 | 12 | 50% |
| 21-30 | 0 | 0 |
| 31-40 | 2 | 8% |
| 41-50 | 3 | 13% |
| 51-60 | 1 | 4% |
| 61-70 | 0 | 0 |
| 71-80 | 0 | 0 |
| 81-90 | 0 | 0 |
| >90 | 0 | 0 |
| Prefer not to say | 0 | 0 |

**Table 32**: Gender

|  | N | % |
|---|---|---|
| Male | 23 | 96% |
| Female | 1 | 4% |
| Other | 0 | 0 |
| Prefer not to say | 0 | 0 |

Table 33: Highest level of education

|  | N | % |
|---|---|---|
| Middle school | 1 | 4% |
| High school | 17 | 71% |
| Community College | 0 | 0 |
| 2 Year Degree | 0 | 0 |
| College | 1 | 4% |
| Master's Degree | 4 | 17% |
| Doctorate | 1 | 4% |
| Other | 0 | 0 |
| Prefer not to say | 0 | 0 |

Table 34: Race/Ethnicity (check all that apply)

|  | N | % |
|---|---|---|
| American Indian or Alaska Native | 1 | 4% |
| Asian | 4 | 17% |
| Black | 0 | 0 |
| Hispanic or Latino | 7 | 29% |
| Native Hawaiian or Other Pacific Islander | 0 | 0 |
| White | 17 | 71% |
| Some other race or ethnicity | 1 | 4% |
| Prefer not to say | 0 | 0 |


**Ethical Statement**

The research in this study is novel, representing the authors' analysis, experience, and perspectives. We had no intent to cause harm to others. The pre- and post- questionnaires were managed by the program evaluator, Technology for Learning Consortium,

and used as part of course material. Care was taken to keep responses anonymous. The Florida Atlantic University Social, Behavioral and Educational Research IRB determined that this program did not meet the definition of human subjects research according to federal regulations (24 March 2021). Therefore, it was not under the purview of an IRB.

**Competing interests**

The contact author has declared that none of the authors has any competing interests.

**Acknowledgements**

We thank all community members that took the time to listen to our youth voices through Climate READY. Thank you to all high school/dual enrolled participant Climate READY Ambassadors from around Palm Beach County, and the participant fourth and fifth grade after-school students, teachers, and after-school staff at Barton Elementary School in Lake Worth Beach, Coral Sunset Elementary School in Boca Raton, Galaxy E3 Elementary School in Boynton Beach, Pine Jog Elementary School in West Palm Beach, and the University Learning Academy in Riveria Beach. We thank all program partners including the School District of Palm Beach County, Galaxy E3 Elementary School, the Palm Beach County Office of Resilience, the Coastal Resilience Partnership, the Southeast Florida Regional Climate Change Compact, and city sustainability offices in Boynton Beach, Boca Raton, Delray Beach, and West Palm Beach. We also thank Earth Force, the South Florida Science Center and Aquarium, Boca Raton Community High School, FAU Center for Environmental Studies, FAU Department of Educational Leadership and Research Methodology, MANG nursery, Grassy Waters Preserve, Page Turner Adventures, Inc., Boca Save our Beaches, Delaney Reynolds of the Sink or Swim Project, Barbara Riley for your mentorship in environmental science storybook collaborations with elementary school teachers, FXB Climate Advocates, and our CR Advisory Council for providing time and effort towards our program and contributing to its success. We also thank our EGUsphere peer reviewers Eleanor Burke, Jan Cincera, and William Finnegan, and guest editor David Crookall for your insightful suggestions and efforts to help us make this a manuscript readable for a diverse audience. Finally, we thank our funding source, the Environmental Literacy Program with NOAA Education, for giving us the opportunity to design and implement Climate READY.

**Financial Support**

This study was prepared by FAU Pine Jog Environmental Education Center under award NOAA-SEC-OED-2020-2006190 from the Environmental Literacy Program of the National Oceanic and Atmospheric Administration (NOAA), U.S. Department of Commerce. The statements, findings, conclusions, and recommendations are those of the authors and do not necessarily reflect the views of NOAA or the U.S. Department of Commerce.

**Data Availability**

The authors confirm that data collected for this study support their findings and are found within the article and in supplementary materials.

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
