# Peer review of "Assessment of a youth, climate empowerment program: Climate READY"

_EGUsphere, 2024_

## Author Response (AR1)

**RC1**: 'Comment on egusphere-2024-139', Eleanor Burke, 13 Feb 2024
Climate READY: A three-semester youth empowerment program

**General Comments**

This article describes a 3-semester credit-bearing program called "Climate READY" at Florida Atlantic University's satellite campus, Pine Jog Environmental Education Center (FAU Pine Jog). It is well organized and well written, with clearly stated project objectives and a testable and measurable hypothesis that the program would "increase the environmental literacy of 4–12 grade students in Palm Beach County, FL and the general community that they live in so that they can become more resilient to extreme weather and/or other environmental hazards, thus empowering them to become involved in achieving that resilience."

The article describes the program in depth: the methods section is written with thorough detail; the results section provides a great deal of data from pre- and post-testing of both the high school students in the dual-enrollment (high school/college credits) cohort, and of their younger 4th-5th grade mentees in an after-school program on the FAU Pine Jog campus. Ample and clear data tables register both groups' learning of basic facts about climate-change, as well as their attitudes and sense of perceived personal agency around climate change readiness in their communities. Data show how community outreach service presentations by the Ambassadors changed their sense of community connection and personal agency. The data and statistical analyses support the discussion and conclusions the authors draw from it.

The project and this article fall nicely within the mission and scope of GC, i.e. providing "outreach, public engagement, widening participation, knowledge exchange, and…any initiative which seeks to communicate an aspect of geoscience to a wider audience than the experts within that particular field." The project itself focused on exactly these goals, and this article is written for accessibility to a diverse audience, which might include primary and secondary school teachers and districts, college and dual-enrollment environmental program planners, local groups and governing boards who consider climate change readiness at the community level, local/regional environmentalist groups, parents and home-schooling families, school-community partnerships, and others.

Some new or lesser-known concepts and tools featured in the narrative include the following:

- An after-school/college partnership with a three-pronged programmatic approach to teaching climate literacy and climate-change readiness: (1) a dual-enrollment college course for high school students, (2) a cross-age peer teaching/learning experience, and (3) a community outreach service program, all based in the 3-semester college course using original course curriculum designed by the authors.
- Ample data points from pre- and post- testing (of both age-groups) using questions designed by the authors, many of which are based on high school and college earth science textbooks, and on the Climate Literacy and Energy Awareness Network's *Climate Literacy Quiz*.
- Use of the NOAA-created earth-science technology tool, SOS–*Science on a Sphere,* to teach both age groups, including for the cross-age peer teaching.
- Measured evaluation of the "train the trainer" technique, with high schoolers as the mentors for younger students and also as presenters for community groups, events, and boards.

All of these features merit the documentation by this article. It was a pleasure to read. EB

**Specific Comments**

**Title** Concise and to the point. I had some confusion over which age-group of "youth" was your focus for "empowerment", then realized it was both. You might want to clarify early in the intro.

**Abstract**

-Throughout the article there are frequent shifts back and forth from grades 4-5 mentees to CR ambassadors (e.g. L242 "students that participated in the program"--mentees? Ambassadors too?), which confuses the reader when both groups are sometimes referred to as students. Can you find a shorthand way to distinguish between age groups when speaking about "students?" The use of "Ambassadors" works for older ones; you need a term for their mentees.

**Intro**

-L43-50–*Without some kind of professional development, teachers are unsure how to approach these topics.* Yes, as Lambert found in 2012, teacher-preparation issue was a major factor then, but it's now 12 years later, and teachers need to be given credit for the significant professional development most have taken since then. Let's move away from blaming teacher-prep. Can you acknowledge Florida's unfriendly soil for climate science education standards, especially in light of the state's Stop-W.O.K.E. act, and the teachers' legitimate fear of losing their jobs if they teach topics or use vocabulary some parent or community member might object to? How to work around this conundrum?

–Also might mention - where FL lines up with other states' standards about Climate Change,e.g. https://ncse.ngo/making-grade-how-state-public-school-standards-address-climate-change , which studied all 50 states' science standards w/ respect to climate change info/education. I understand that these may be touchy areas, but we have free academic speech in this country, and it is so important to exercise it–especially in universities!

-L70 A few places of ameri-centrism to fix, e.g. *next environmental "ground zero"*--specify "in US" or region; check throughout for USA-centric references and try to relate them in global terms

-Consider a side-note or endnote near the Abstract, explaining (for non-American readers) terms like Title 1, free & reduced lunch, dual enrollment (line 200 is too late). Also ages of students in grades 9-12 and 4-5, early in the Intro.

**Methods**

–L150– First paragraph is confusing. Maybe bullet list the 3 components. Tighten it up.

–So sad that Covid interrupted the first year of a beautifully planned program. You made lemonade from the lemons.

–L175-200 It's a little confusing about how (and why) the Galaxy E3 students were part of the high school students' learning about CC with SOS, and whether these overlap with the after-school program students who were mentored by the high schoolers. After several reads it gets clearer, but few folks will read 3 times. Please tighten this up.

**Results**

All quite clear except -L335 one misstatement– "fewer" should say more, if I'm reading correctly. Table 7 bears this out. *After their experience, ...significantly **fewer** students reported that they do not question [ADD: **the science of**] climate change (Table 7)* The sort-of double negative implied by the question makes it pretty confusing.

–L327 *Most students (96%) were between 15 and 16 years of age and female (72%), 55% of the students described themselves as White, 36% as Hispanic or Latino, and 18% as Black.*

Did this reflect local populations? Seems it might be skewed white? Given that *"Students from underserved communities were prioritized in the recruitment process"* (L200), was this

disappointing? Maybe treat this question in the Discussion section?  Is this the impetus for your recommendation to do multiple in-person recruitment sessions?

**Discussion section**

–The fourth feature discussed in general comments above, training high schoolers as "ambassadors" to do dispersed outreach to elder and younger counterparts in their communities, might gain more credence from skeptics if supported with some research citations about the efficacy of cross-age peer teaching and of community service-learning.

[Some places to start might include:

- Promoting Positive Youth Development Through Teenagers-as-Teachers Programs,   Steven M. Worker, Iaccopucci, Bird, and Horowitz, *Jl Adolescent research*  (2018)  https://doi.org/10.1177/0743558418764089
- Filges T, Dietrichson J, Viinholt BCA, Dalgaard NT. Service learning for improving academic success in students in grade K to 12: A systematic review. Campbell Syst Rev. 2022 Jan 7;18(1):e1210. doi: 10.1002/cl2.1210. PMID: 36913211; PMCID: PMC8741202.]

–You might make a little space for connecting this project to the wider world of similar collaborations between high schoolers (as mentors, teachers), and younger students? And for highlighting how you see your project developing and perhaps disseminating, being replicated in other districts and regions.

--And where your students' pre/post beliefs line up with USA  (and wider world?) (see EdWeek Research Center survey, October 2022  https://www.edweek.org/teaching-learning/teens-know-climate-change-is-real-they-want-schools-to-teach-more-about-it/2022/11

–And more clearly state your own novel contribution to the literature on Climate Change Readiness education, (such as the curriculum you developed, and train-the- trainer/peer teaching approach.)

**Conclusions section**

–Could use a statement about future programmatic or research steps, if any are planned. Suggestion for further research by FAU and others, would be whether school-college partnerships and cross-age peer teaching could serve to scale up community/regional climate readiness quickly.

–As an aside, I presume you got parental permission for public presentations for the ambassadors under the age of majority? Could be important, especially if there is a community member present with a political ax to grind. This happened to our high school students while presenting at a town meeting. We (staff) stepped in to stop the speaker from becoming abusive, and the high school students were unfazed, but parental permission was a safety-net throughout our students' public presentations road-show. You might do well to include it as a recommendation for other school-college partnerships tracing your footsteps, which I hope they will do.

**Technical corrections**

L 6–spell check for dropped "d" on "Dedicate" Youth–might be in a couple of places

L 21 Data were collected from students—specify both age groups

L.34. minimal level of environmental literacy that is, the possession of...   awk. punctuation

L 54 grades 4-12 (youth ages 9 to 18)    specify 2 groups, 9-10 and 14-18?

L 174 heading is about High School ambassador program, but next para goes into SOS curriculum developed in collaboration with E3 Galaxy elem. School–confusing at first

–Paragraphs at L175 and L199–might it help to switch their order, to give a better picture of the Ambassador program before delving into the SOS curriculum and Galaxy school collaboration they followed?

Table 18–your students' results may have been affected by the wording of the question. Since *current* sea level rise is not specified in the question, other answers could be marked as correct, since future rise may well occur with ice sheet melting, etc.

-L490 table 23, why are there check marks on "Shorter days & fewer hurricanes"? These would not be considered correct answers, thus should not be checked, right?

**Citation**: https://doi.org/10.5194/egusphere-2024-139-RC1

Response

Eleanor,

Thank you for your detailed review of our article. It was truly informative and we are working through the manuscript to make several of the edits you have suggested. We've included replies below.

Title – We evaluated the term "Youth" used in the title and Intro. We've defined it in L41 and L54. Perhaps this is sufficient?

Abstract – We will use "ambassadors" for teens and "after school students" for the mentees that are 4th and 5th graders.

Intro – We want/need to stay neutral with Florida politics, so we don't want to point out any potential misgivings that may seem obvious to others. However, we see your point in giving teachers credit. There is evidence that more resources are available, but the majority of teachers that we have worked within our area are already swamped with additional procedures and paperwork from local and state levels, leaving little time to do anything extra or outside of local and state requirements. Most classrooms in our area are driven by state standards and the frequency to which these standards are tested on.

Florida was rated a "D" in the NCSE study. We were aware of this and have cited it in the past. Thank you for pointing this out. We will add this reference and include a mention.

The term "ground zero" has been used many times in our community and in publications. We cited one of them, however we can add a sentence which elaborates on its meaning.

In the abstract, we opted to use the term "low socio-economic communities" (L10) in place of Title 1, etc. We will evaluate the two terms. Perhaps mentioning Title 1 with low… in the abstract would help? The range of ages of the children are in mentioned in L41 and L54, however we can list the ages more specifically to grade levels if needed.

Methods – We will bullet point the course references to tighten that up (L150). It will read better. Covid was a mess for everyone, thank you for acknowledging our efforts to move forward regardless the hardship it brought. As for L175-200, Galaxy Elementary received a science on a

sphere installation in 2013. It was the first elementary school to get one and in a low socio-economic community, making it very unique. We will clarify that in the manuscript.

Results – The phrasing in L335 is confusing. We've debated how it's worded before submitting the article for review. Your observation clearly indicates that it needs revised to take out the double negative. As for L200, "underserved communities" is not limited to race, it also includes populations in poverty, which our study targets – "low socio-economic communities." More than 80% of the students that participated in Climate READY were from these communities. With that said, the US Census Burau reports that the population of Palm Beach County is represented by 52.3% white, 17.1% black, 23.5% Hispanic/Latino, and the remaining 7.1% as various. For the exception of the Hispanic/Latino population (36%), these values are not far off from the study. Upon further investigation, the population demographics of students within the Palm Beach School District, a public school system, is 29% white, 27% black, and 36.9% Hispanic/Latino. This gives us some things to think about. We will consider these items while revising this section. In some ways it seems representative of Palm Beach County as a whole, but perhaps skewed when looking at the population in our public school system.

Discussion – Thank you for bringing additional papers and the EdWeek survey to our attention. We will review these and see where they may fit in our discussion. We're also considering adding a section that at least highlights our curriculum and train-the-trainer/peer teaching approach.

Conclusions – There are a few suggestions that we could add to continue our work or expand to other regions, etc. We've applied for funding to reach out to surrounding counties in South Florida, but we haven't been successful in securing those funds (yet).

Thank you for pointing out parental permissions. It's very important to us and to the university as a whole to protect our students, especially minors. All students and their families were given forms to complete to acknowledge the potential for sharing information about the program. Thankfully, this is what our legal department does best!

Technical Corrections – Great eye on mistakes and typos! Most have been noted and will be fixed, especially the check marks on Table 23. We didn't realize that. We will also consider the 2 paragraphs within L175 to L199. It might be a matter of rewording. Lastly, Table 18 only has two correct answers, "melting glaciers and ice sheets" (missing checkmark) and "seawater expanding as it gets warmer." There are several references for this, but here is a quick one from NOAA where they state "The two major causes of global sea level rise are thermal expansion caused by warming of the ocean (since water expands as it warms) and increased melting of land-based ice, such as glaciers and ice sheets."
https://oceanservice.noaa.gov/facts/sealevel.html#:~:text=The%20two%20major%20causes%20of,as%20glaciers%20and%20ice%20sheets.
Sea ice does not affect sea level and accelerated rivers could be a confusing option, but it was never discussed in class.

Thank you for your valuable time to provide us with this feedback. It will definitely help in making this paper shareable and readable for a diverse audience.

All the best, RW
* * *
**CC1**: 'Comment on egusphere-2024-139', Jan Cincera, 05 Apr 2024
The paper describes an exciting youth CCE program. I appreciate that the program is clearly described with a lot of details, and I also appreciate the overall quality of the program.

At the same time, I suggest improving the paper to enhance its quality as a scientific paper. Providing a more detailed background would situate the Climate READY program within the broader context of environmental education and youth empowerment initiatives. This could include reviewing similar programs, their outcomes, and how Climate READY fills existing gaps.

Secondly, the paper needs to be adequately related to other existing literature. Strengthening the literature review by discussing theoretical frameworks would enhance its overall quality.

Furthermore, a more detailed explanation of the data collection and analysis methodologies would be helpful. When an adapted instrument is applied (CNS), its source should be cited appropriately.

The presentation of the findings could benefit from more clarity. Consider moving some of the unnecessary tables to the Appendix and present just the most important (significant) findings. Regarding the low number of respondents, I suggest being careful in its interpretation. Qualitative data (Table 19) should be categorized. Consider moving the table to the Appendix and presenting just the most critical findings in the text.

Generally, you should not present new findings in the Discussion section (e.g., Fig. 6 should be presented in Findings). Instead, consider concisely synthesizing the main findings and relating them to the relevant literature or similar CCE programs in the Introduction. In addition, you can provide concrete suggestions for educators, policymakers, and community organizers based on the findings and discuss how the program's strategies can be adapted to different contexts or scaled to reach broader audiences.

Although I suggest many changes, I want to highlight the quality of the presented program and the need for its presentation to the public.
Good luck with your paper!
Jan

**Citation**: https://doi.org/10.5194/egusphere-2024-139-CC1

Jan,

Thank you for taking the time to read through our manuscript and providing valuable feedback. We are planning to include a small section that provides a bit of background on climate education and youth empowerment programs. We consider ourselves as part of a "community of practice," something that NOAA has fostered over the years, so there are a few programs that we can cite that have contributed to the conversation.

It's unclear what you perceive as missing in the data collection and analysis sections. We've cited the evaluation tool starting on L285, described and cited how pre-post surveys were conducted starting on L293, and provided a description of our data analysis starting on L309. Unsure of what you mean by "CNS."

In regards to Table 19, we see value in sharing the student free response answers as they wrote them, but we also see how it can be overwhelming. To address this, we created Table 27 in the Appendix that shows a ranked tally of the responses. In light of your observation, perhaps switching these would be beneficial, moving the ranked tally to the results section and the student responses to the appendix?

We were not considering Figure 6, the word cloud, as part of the data analysis. It's not part of our methods, se we didn't include it in results. However, we wanted to use it while discussing the results as it was an artifact of the program. We'll discuss this and consider your comment about moving it to results.

Another reviewer mentioned our lack of suggestions for readers/educators, so we plan to address that and add to our conclusions.

Thank you for your time. Your observations are insightful and we will consider them moving forward.
Best,
RW
* * *
**RC2**: 'Comment on egusphere-2024-139', William Finnegan, 15 May 2024

I enjoyed learning about this program and reading this preprint. In general, I think this is a really interesting initiative, and it makes sense to write up your evaluation for this publication. However, I think it needs major revisions, mostly in the form of restructuring, consolidating redundant sections, and adding a few missing pieces.

General Comments:

This article provides a very detailed evaluation of a climate education intiative in Florida. There are a number of interesting and novel features of this program that are clearly described and studied. However, my main feedback would be to revisit the article so that it is less of a program evaluation, and more of a contribution to the growing body of literature related to climate change education. With that in mind, I would encourage you to add more of a literature review section to the introduction, especially in terms of concepts of environmental literacy and relevant

educational theories with respect to the pedagogical (e.g., place-based, active learning, case studies, Photovoice) and methodological approaches of both the program and research. Similarly, towards the end of the article you very much focus on specific operational recommendations rather than reflect on the results of your research and how this could be of use to other researchers and practioners in terms of pedagogy/methodology.

The desciptions of the program are combined with the descriptions of the data collection tools used in the research/evaluation. I think the article would benefit from breaking these out more clearly into distinct sections. For example, Section 2.2 could be a new section 3 focused on Research Methodology, and Section 2.1 could be combined with some of the related/redundant program description in Section 4. This restructuring would also allow you to refocus Section 4 on discussing the results in Section 3 in relationship to the ideas introduced earlier in the article, and advance the contributions of this study related to environmental education research and practice, rather than simply describing the program activities in relation to the program objectives.

You include a great amount of detail in the results section (2), and include a 10-page appendix of additional information (that lacks a clear focus or structure). While this level of detail makes your research very transparent, it also makes it difficult for the reader to easily identify the key findings and contributions of the article. I would suggest revisiting the results and appendix with an eye on the findings most related to your hypothesis. It might make sense to create multiple appendices with a clearer focus (Teaching Materials, Program Outputs, etc). As an example of too much detail for a research article (versus program evaluation), I don't think you should include Table 19 in the body of the article, but could consider including in an appendix.

At the same time, there are a some aspects of the program which I think may be underutilised in terms educational interventions that result in program outputs that could also serve as data for qualitative analysis (beyond the quantitative data of the pre/post tests). You mention Photovoice in passing, which is a well established participatory action research method. The storybooks are also a really interesting aspect of the ambassadors' engagement with younger students. While you reflect of the value of storytelling, and include some more information in the appendix, there is no detailed explanation of the specific themes, narratives and creative approaches in the storybooks. To me, this is the most interesting part of your project (students teaching students, storytelling supporting climate resilience, etc).

Specific Comments:

In the introduction, I think it would make sense to include a citation for the definition of environmental literacy outlined.

When you refer to grades 4-12 (ages 9-18) in section 1.1, it raises a question of the appropriateness of such a large age range for an educational intervention. Later in the article you make clear there are two distinct populations: the primary participants (high school students) and the seconday participants (grade 4-5), which makes more sense. I would suggest you explain the specific audiences and ages in more detail in 1.1, perhaps simply as a parenthetical.

In Figure 1, I don't think the second map (b) adds much value, especially given the similar scale and more stylised icons and edges.

In section 4.1.4, are you introducing a new community survey that hasn't be mentioned earlier in the article? I wouldn't mention an additional form of data collection for the first time in the discussion section, so definitely worth describing this earlier (the survey in the methods section, and the responses in the results section).

**Citation**: https://doi.org/10.5194/egusphere-2024-139-RC2

William,
Thank you for taking the time to read through our manuscript and provide valuable feedback.

This particular manuscript highlights one year of the 3-year project, so though it may appear as an evaluation it is not a full evaluation. Information on our full evaluation of the program was shared with our grant provider, NOAA Education in January 2024. Therefore, the methods and data shared in this particular manuscript are only reflecting the single completed year that was not affected by COVID or the restrictions that the pandemic caused while implementing the 3-semester course model. Comparing that data would be a much larger paper than it already is. We were obviously very ambitious while planning for this project and part of what we want to share in this manuscript are the lessons learned while taking our initial visions and turning them into a successful program while going through unexpected changes.

After reading all of the reviewer comments, including yours, we are planning to include a small section that provides a bit of background on climate education and youth empowerment programs, which should add to literature review of related materials. As with other reviewer remarks, we have noted that our recommendation in this manuscript targets our own operational improvements, but it lacks recommendations for others. Thank you for pointing that out. We are planning to elaborate on the statement that "It also provides an example of a youth empowerment program that could be shared and implemented within other colleges and universities" (L728).

While writing this manuscript we were following similar formats from other EGU Geoscience Communication published papers such as "Using paired teaching for earthquake education in schools" (Solmaz et al 2021). However, your suggestions to restructure the order of our sections (2.1, 2.2, 3, 4) has given us some thought. We will consider your comments.

The idea of multiple appendices is interesting. Is this common with other Geoscience Communication articles? We'll revisit this idea with the editors. The topic of Table 19 has come up before, so we are planning to change that. We've provided a response to another reviewer to make the information more user friendly. Thank you for noticing.

We appreciate your interest in the storybook narratives and creative approaches. There is enough information from that one lesson that could result in an additional manuscript. However, it's not necessarily the focus of this paper, as the focus is the 3-semester model, which includes these lessons (storybook and photovoice). We chose to focus on storytelling in this paper because both lessons and the accompanying activities/lessons have a common theme of storytelling, which to us is the most important aspect of the program. Ultimately, our students become storytellers using photos, writing original stories, and practicing communication skills including public speaking while expressing their knowledge and concern about climate change. We will consider making this clearer in our discussion. Thank you for that feedback.

Introduction – We are planning to add more citations to support environmental literacy.

Section 1.1/grades 4-12 – We have plans to make the age ranges/groups clearer based on yours and other reviewer's observations.

Figure 1 – The second map was part of an artifact of the program that we used on multiple occasions within our own communities. This map is much easier to read and helps to understand the location of the majority of the distinct communities we served (marked with a star). The first map is a greater view of Palm Beach County and an effort to show context as well as show the location of the Pahokee/Belle Glade agricultural area that we reference. It is relatively far removed from the immediate coastal region, but in some ways the area is more vulnerable to environmental change and a very important part of the county. It's often overlooked and underrepresented. We see value in both maps. Would it help to provide more explanation in the manuscript?

Section 4.1.4 – The community survey was only optional for community members and participation was not formal, so we were not planning to include this in our methods and results. Instead, we opted to use the information as an artifact of the program, something that was not expected, and thus use it to help us tell the story of our 3-semester dual enrollment program. This is why we placed it in the Discussion section.

Thank you for your review. You have given us some things to consider that will make it a better manuscript.

Best,
Rachel

**EC1**: 'Lead guest ed comment on egusphere-2024-139', David Crookall, 11 Jun 2024

DC(ed) comments, by David Crookall, after reviews and responses – 11 June, 2024
**egusphere-2024-139 – Climate READY**

My recommendation to the Exec Eds of GC is that, after full revision, this ms will be ready for publication.

I have (re-)read the reviewers' comments and the author's responses carefully. I wish to thank the three reviewers, Eleanor, Jan and William, for the time and trouble that they took to review Rachel et al's ms. The reviews were insightful, constructive and relevant. Rachel's three responses indicated a real desire to incorporate most of the reviewers' suggestions. A few of the suggestions, although of real interest, seemed to go a little beyond the scope or purview of the objectives or intentions of the ms; and the author has clearly justified in each case their hesitancy to expand an already large (and somewhat complex) ms.

In light of this, further, long remarks from me might complicate things further and make the heavy task of revising the ms too burdensome. A few quick comments might help, however:
- **Appendices**. Yes, several appendices are welcome. Give each a N° and a name, and refer to them in the body of the ms.

- **Complexity & scope**. Avoid adding stuff that is likely to complicate the central issues, points and objectives. If necessary, include it in an appendix.
- **Structure**. As William, and the other reviewers in their own way, suggested, restructure the body of the ms, so that sections are clear, with headings, and have subsections with clear subheads. Check that each section/subsection has a short introductory para to intro the section. Incl a topic sentence for each para or for most paras.
- **Objectives**. It would probably help the reader if you compose two sets of objectives: one set for the article as a whole (maybe call them goals), and another set for the research (research objectives).
- **Lit**. You might wish to differentiate two aspects of your lit review. One that focusses on your research objectives. (You research objectives will, of course, have been driven more by your program than by gaps in the lit.) One or CCE in general, in order to help the ready see the educational and academic context of your article. I am not sure how feasible this is.
- **Terminology**. To help a wider, international audience, it is probably worth doing a glossary of terms. Fort example: school levels, title 1, literacy v education, students v ambassadors, etc. You could put this early on in the body of your ms, or in an early appendix, and refer to it in the body.
- **Writing**. Please refer closely to these guides as you revise:
  - https://oceansclimate.wixsite.com/oceansclimate/author-guide
  - https://oceansclimate.wixsite.com/oceansclimate/writing-guide
  - https://oceansclimate.wixsite.com/oceansclimate/there
- **Title**. I would suggest a more focussed title, as follows:

**Assessment of a youth, climate empowerment program: Climate READY**

  The number of semesters is a small point for the title of a large study.

- **Audience**. Please remember that your audience is far wider than your CCE colleagues, and extends to the whole readership of GC, that is, the 'whole wide world' J

I am sure that I have missed some important things, so, if you have any questions, please do not hesitate to contact me. I will be away, with only light and sporadic internet access from 21 June until 27 July, 2024.
Good luck – happy revising, david
**Citation**: https://doi.org/10.5194/egusphere-2024-139-EC1